# CONSISTENCY MODEL IS AN EFFECTIVE POSTERIOR SAMPLE APPROXIMATION FOR DIFFUSION INVERSE SOLVERS

## ABSTRACT

Diffusion Inverse Solvers (DIS) are designed to sample from the conditional distribution with a pre-trained diffusion model an operator and a measurement derived from an unknown image. Existing DIS estimate the conditional score function by evaluating operator with an approximated posterior sample. However, most prior approximations rely on the posterior means, which may not lie in the support of the image distribution and diverge from the appearance of genuine images. Such out-of-support samples may significantly degrade the performance of the operator, particularly when it is a neural network. In this paper, we introduces a novel approach for posterior approximation that guarantees to generate valid samples within the support of the image distribution, and also enhances the compatibility with neural network-based operators. We first demonstrate that the solution of the Probability Flow Ordinary Differential Equation (PF-ODE) yields an effective posterior sample with high probability. Based on this observation, we adopt the Consistency Model (CM), which is distilled from PF-ODE, for posterior sampling. Through extensive experiments, we show that our proposed method for posterior sample approximation substantially enhance the effectiveness of DIS for neural network measurement operators (e.g., in semantic segmentation). The source code is provided in the supplementary material.

## 1 INTRODUCTION

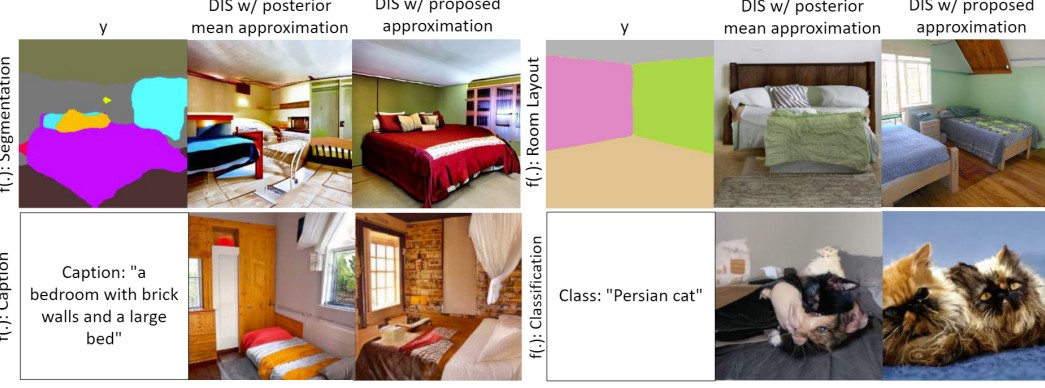

Figure 1: A visual comparison of DIS with posterior mean as approximation for posterior sample, and DIS with proposed CM approximation for posterior sample.

Diffusion Inverse Solvers (DIS) are a family of algorithms designed to address the inverse problem using diffusion priors (Li et al., 2023; Moser et al., 2024). Specifically, given an operator $f(.)$, a measurement $y = f(x_0')$ from some unknown image $x_0'$, and a pre-trained diffusion model $p_\theta(X_0)$, DIS aims to sample from the conditional distribution $X_0 \sim p_\theta(X_0|y)$. For example, when $f(.)$ is a down-sampling operator, DIS functions as a perceptual super-resolution algorithm (Menon et al., 2020); when $f(.)$ is an image segmentation operator, DIS generates an image which has the same

segmentation map as the given one (Ye et al., 2024; Bansal et al., 2023) (See Figure 1). However, directly sampling from conditional distribution $p_\theta(X_0|y)$ is intractable. To tackle this challenge, previous works have adopted a variety of techniques such as linear projection (Wang et al., 2022; Kawar et al., 2022; Chung et al., 2022b; Lugmayr et al., 2022; Song et al., 2022; Pokle et al., 2024; Cardoso et al., 2024), variational inference (Feng et al., 2023; Mardani et al., 2023; Janati et al., 2024), Bayesian filtering (Dou & Song, 2023), sequential Monte Carlo (Wu et al., 2024a; Phillips et al., 2024), proximal gradient methods (Xu & Chi, 2024) and conditional score estimation (Chung et al., 2022a; Yu et al., 2023; Zhu et al., 2023; He et al., 2024; Song et al., 2023b; Boys et al., 2023; Rout et al., 2023; 2024).

Among various DIS techniques, conditional score estimation methods (Chung et al., 2022a; Song et al., 2023c) are the most widely adopted. These methods are suitable for general non-linear operators $f(.)$ and are practically efficient. During the diffusion process from $X_T$ to $X_0$, they estimate the conditional score $\nabla_x \log p_\theta(X_t|y)$ with posterior samples from $p_\theta(X_0|X_t)$. Since the posterior $p_\theta(X_0|X_t)$ is intractable, various approximations to posterior sampling are proposed based on the posterior mean. These approximations either directly adopt the posterior mean (Chung et al., 2022a; Yu et al., 2023; Zhu et al., 2023; He et al., 2024) or construct a Gaussian distribution centered at the posterior mean (Song et al., 2023b; Boys et al., 2023; Rout et al., 2023; 2024).

However, these posterior mean-based approximations often yield posterior samples that are far from real images. It is well-known that the mean of a set of noisy images may not lie within the support of the image distribution (Ledig et al., 2017; Blau & Michaeli, 2018; Zhang et al., 2024). While these out-of-distribution approximations have been shown to be successful for simple operators $f(.)$ such as down-sampling and motion blurring, they may fail for more complex operators, especially when $f(.)$ are neural networks such as segmentation or classification (Ye et al., 2024; Bansal et al., 2023). This is partially because neural networks are particularly sensitive to out-of-distribution inputs, which can significantly degrade their performance. On the other hand, consistency models (CM) (Song et al., 2023c) can effectively produce high quality images with a few steps, while their relationship to the posterior samples is under-explored.

To address the aforementioned challenges, we propose the following points:

- We propose a novel approach for approximating posterior samples in DIS. Our approximations are guaranteed to be valid images and perform well with neural network measurement operators.

- To justify our approximations are effective posterior samples, we first intuitions and examples on why the solution of the Probability Flow Ordinary Differential Equation (PF-ODE) might have high density in posterior. Next, we demonstrate that the solution of PF-ODE has positive density lowerbound in posterior with high probability.

- Empirical results show that utilizing CM for posterior sample approximation significantly improves the performance of DIS, particularly when the operators are neural networks, such as semantic segmentation and image captioning.

## 2 PRELIMINARIES

**Diffusion Model** The diffusion model is a type of generative model consisting of a $T$-step Gaussian Markov chain in continuous space (Sohl-Dickstein et al., 2015). Two widely adopted diffusion models are the variance-preserving (VP) diffusion and variance-exploding (VE) diffusion models. In this work, we follow the VE diffusion formulation (Song et al., 2020); for the details on the VP diffusion, refer to Ho et al. (2020); Kingma et al. (2021). We denote the source image as $X_0$, and the forward process of VE diffusion can be described as a Markov chain:

$$q(X_T, ..., X_1|X_0) = \prod_{t=1}^{T} q(X_t|X_{t-1}), \text{ where } q(X_t|X_{t-1}) = \mathcal{N}(X_{t-1}, (\sigma_t^2 - \sigma_{t-1}^2)I), \quad (1)$$

where $\sigma_t^2$ are hyper-parameters commonly referred to as the variance schedule. The reverse diffusion process is also a Markov chain, with the transition kernel $p(X_{t-1}|X_t)$ dependent on the score

function $\nabla_{X_t} \log p(X_t)$, which is approximated by a neural network $s_\theta(t, X_t)$ parameterized by $\theta$:

$$p_\theta(X_0, ..., X_T) = p(X_T) \prod_{t=1}^{T} p_\theta(X_{t-1}|X_t),$$

$$\text{where } p_\theta(X_{t-1}|X_t) = \mathcal{N}(X_t + (\sigma_t^2 - \sigma_{t-1}^2)s_\theta(t, X_t), (\sigma_{t-1}^2/\sigma_t^2)(\sigma_t^2 - \sigma_{t-1}^2)I). \tag{2}$$

Song et al. (2020) demonstrated that the reverse diffusion process can be viewed as the discretization of a reverse stochastic differential equation (SDE) (Anderson, 1982). Furthermore, they introduced the Probability Flow Ordinary Differential Equation (PF-ODE), which shares the same marginal distribution $p_\theta(X_t)$ as the reverse SDE:

$$\text{reverse SDE: } dX_t = -\frac{d\sigma_t^2}{d_t} s_\theta(t, X_t)dt + \sqrt{\frac{d\sigma_t^2}{d_t}} dB_t \quad \overset{\text{same } p_\theta(X_t)}{\Longleftrightarrow}$$

$$\text{PF-ODE: } dX_t = -\frac{1}{2}\frac{d\sigma_t^2}{dt} s_\theta(t, X_t)dt, \tag{3}$$

where $B_t$ is the standard Brownian motion. Therefore, solving either the reverse SDE or the PF-ODE is equivalent to sampling from the reverse diffusion process. Another useful result is Tweedie's formula (Efron, 2011), which offers an efficient estimation of the mean of the posterior $p_\theta(X_0|X_t)$:

$$\mathbb{E}[X_0|X_t] = X_t + \sigma_t^2 s_\theta(t, X_t). \tag{4}$$

**Diffusion Inverse Solvers with Conditional Score Estimation** Given an operator $f(.)$, a target measurement $y = f(x_0')$ from an unknown $x_0'$, and a diffusion model $p_\theta(X_0)$, Diffusion Inverse Solvers (DIS) aim to sample from the conditional distribution $p_\theta(X_0|y)$. In this paper, we focus on DIS that utilize conditional score estimation (Chung et al., 2022a; Song et al., 2023b). Specifically, this paradigm of DIS attempts to estimate the conditional score $\nabla_{X_t} \log p_\theta(X_t|y)$. With this conditional score available, sampling from conditional distribution $p_\theta(X_0|y)$ becomes straightforward by solving the reverse SDE or the PF-ODE in Eq. 3, replacing $s_\theta(t, X_t)$ with the conditional score.

More specifically, Chung et al. (2022a); Song et al. (2023c) propose to decompose the conditional score into an unconditional score and an additional term related to the distance $\Delta(f(x_{0|t}), y)$, where $x_{0|t}$ is a sample from the posterior $p_\theta(X_0|X_t)$ and $\Delta(., .)$ represents a distance metric:

$$\nabla_{X_t} \log p_\theta(X_t|y) = \nabla_{X_t} \log p_\theta(y|X_t) + \nabla_{X_t} \log p_\theta(X_t),$$

$$\nabla_{X_t} \log p_\theta(y|X_t) = \nabla_{X_t} \log \mathbb{E}_{p_\theta(X_0|X_t)}[p(y|X_0)] \approx \nabla_{X_t} \log \frac{1}{K} \sum_{x_{0|t}^{(i)} \sim p_\theta(X_0|X_t)}^{i=1,...,K} p_\theta(y|X_0 = x_{0|t}^{(i)}),$$

$$p_\theta(y|X_0 = x_{0|t}^{(i)}) \propto \exp\left(-\Delta(f(x_{0|t}^{(i)}), y)\right). \tag{5}$$

Under this formulation, an important challenge is how to effectively draw differentiable samples from the posterior $p_\theta(X_0|X_t)$. Direct ancestral sampling from reverse diffusion is computationally expensive. Chung et al. (2022a) propose using the posterior mean computed by Tweedie's formula, as shown in Eq. 4, as the posterior sample. Song et al. (2022; 2023b) suggest modeling the posterior as a Gaussian distribution, with the mean being the posterior mean and the covariance chosen as a hyper-parameter. Rout et al. (2023); Boys et al. (2023) improve upon this by estimating the posterior covariance using the second-order Tweedie's formula. Several other approaches follow the conditional score estimation paradigm and rely on these approximations, including Yu et al. (2023); Chung et al. (2023); Song et al. (2023a); He et al. (2023b); Rout et al. (2024); Meng & Kabashima (2022); Dou & Song (2023); Chung et al. (2022b); Song et al. (2022); He et al. (2023a).

## 3 CONSISTENCY MODEL IS AN EFFECTIVE POSTERIOR SAMPLE APPROXIMATION FOR DIS

### 3.1 PREVIOUS APPROXIMATIONS ARE OUT-OF-DISTRIBUTION

Most previous approximations to $p_\theta(X_0|X_t)$ either directly use the posterior mean or construct a uni-modal distribution centered around the posterior mean. However, the posterior mean $\mathbb{E}[X_0|X_t]$

does not necessarily correspond to a valid image (Ledig et al., 2017; Blau & Michaeli, 2018). In other words, the posterior mean may not lie within the support of the natural image distribution, resulting in its probability density in both marginal and posterior distributions close to zero:

$$p_\theta(X_0 = \mathbb{E}[X_0|X_t = x_t]) \approx 0, p_\theta(X_0 = \mathbb{E}[X_0|X_t = x_t]|X_t = x_t) \approx 0. \tag{6}$$

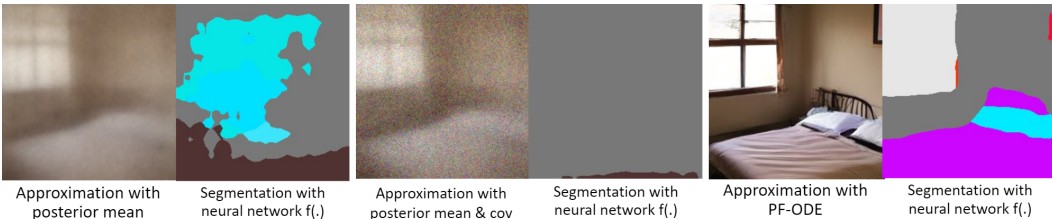

Approximation with posterior mean    Segmentation with neural network f(.)    Approximation with posterior mean & cov    Segmentation with neural network f(.)    Approximation with PF-ODE    Segmentation with neural network f(.)

Figure 2: Different approximations of posterior sample, and their output after a segmentation $f(.)$.

|  |  | FID | KID |
|---|---|---|---|
| Posterior mean | $\mathbb{E}[X_0|X_t]$ | 317.0 | 31e-2 |
| Posterior mean with Gaussian | $\mathcal{N}(\mathbb{E}[X_0|X_t], r_t^2 I)$ | 353.0 | 37e-2 |
| PF-ODE | $\Phi(t, x_t)$ | 24.87 | 1.5e-2 |

Table 1: Sample quality of different approximations.

When the operator is a neural network, such out-of-distribution approximations can significantly degrade the performance of operator. A visual example is shown in Figure 2. The approximations using the posterior mean and those using a Gaussian distribution centered at the posterior mean do not yield valid image samples. Consequently, when these approximations are processed by a semantic segmentation operator $f(.)$, the outputs are nonsensical.

In Table 1, we quantitatively verify that the posterior mean is not a valid sample. We use two widely used metrics, FID (Fréchet inception distance) and KID (Kernel-Inception distance), to measure the quality of the samples. It is shown that the posterior mean and the Gaussian sample centered at posterior mean produce very high FID and KID scores, which indicates a low sample quality.

## 3.2 PF-ODE PROVIDES AN EFFECTIVE POSTERIOR SAMPLE APPROXIMATION

It is known that the PF-ODE and reverse SDE, as defined in Eq. 3, share the same marginal distribution. If the score function is learned perfectly, this marginal distribution of $X_0$ of the reverse process is the same as the original image distribution $p(X_0)$ (Song et al., 2020). Denote the solution of the PF-ODE given the initial condition $X_t = x_t$ as $\Phi(t, x_t)$. This solution lies within the support of the natural image distribution, meaning it represents a valid image with non-zero density:

$$p(X_0 = \Phi(t, x_t)) > 0. \tag{7}$$

Returning to the example in Figure 2, when using the PF-ODE as the posterior sample approximation, the semantic segmentation operator $f(.)$ produces a reasonable result. Furthermore, in Table 1, we quantitatively verify that the result of PF-ODE is a valid image (the sample has low FID and KID scores). However, knowing that $p(X_0 = \Phi(t, x_t)) > 0$ is not sufficient. Since we seek a posterior sample approximation, we need to ensure that the solution $\Phi(t, x_t)$ has positive density in the posterior distribution, *i.e.*, $p(X_0 = \Phi(t, x_t)|X_t = x_t) > 0$.

To the best of our knowledge, the relationship between the PF-ODE's solution $\Phi(t, x_t)$ given the initial value $X_t = x_t$ and the posterior $p(X_0|X_t = x_t)$ is still not well understood. In this section, we provide some intuition and theoretical justification on why the solution of the PF-ODE might have high density in posterior, and demonstrate that the solution of the PF-ODE has non-zero density in posterior with high probability, *i.e.,*

$$p(X_0 = \Phi(t, x_t)|X_t = x_t) > 0 \text{ with high probability (w.r.t. } x_t \sim p(X_t)). \tag{8}$$

**Assumption 3.1.** We assume the following conditions hold:

- The $p(X_0)$ can be approximated by a $d$-dimensional Gaussian Mixture Model (GMM) composed of $N$ Gaussians, each with the same small diagonal covariance $\sigma^2 I$ and mean $\mu^i$:

$$p(X_0) \approx \frac{1}{N} \sum_{i=1}^{N} \mathcal{N}(X_0|\mu^i, \sigma^2 I), \text{where } \sigma < \frac{1}{\sqrt{4\pi e}}. \quad (9)$$

- The solution $\Phi(t, x_t)$ and initial value $x_t$ are bounded, *i.e.*, $\|\Phi(t, x_t)\| < c$ and $\|x_t\| \leq c$ for some constant $c$.

Under this assumption, we can rewrite the PF-ODE using the GMM parameters:

**Lemma 3.2.** *The PF-ODE can be written as:*

$$\frac{dX_t}{dt} = \underbrace{\sum_{i=1}^{N} \frac{w_i}{2} \frac{d\sigma_t^2}{dt} \frac{(X_t - \mu^i)}{\sigma^2 + \sigma_t^2}}_{\text{velocity field } v_t}, w_i = (\exp{(-\frac{\|X_t - \mu^i\|^2}{2(\sigma^2 + \sigma_t^2)})})/(\sum_{j=1}^{N} \exp{(-\frac{\|X_t - \mu^j\|^2}{2(\sigma^2 + \sigma_t^2)})}). \quad (10)$$

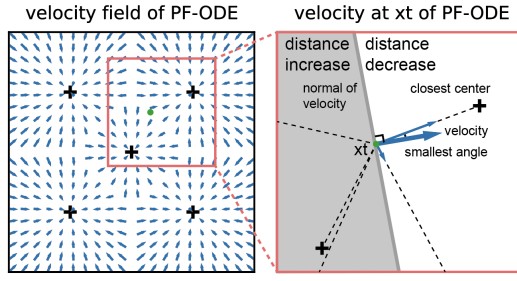

Figure 3: The PF-ODE's velocity field of a 5-GMM.

We can first provide an intuitive analysis of the velocity field $v_t$ of the PF-ODE described in Eq. 10. The velocity field is a sum of vectors pointing towards the centers $\mu^i$, weighted by a softmax function $w_i$. When $\sigma_t^2$ is small, $w_i$ approximates a "hard"-max function that selects the closest center $\mu^*$ to the initial point $x_t$, causing the velocity field to always point towards $\mu^*$. This closest center $\mu^*$ corresponds to the highest density mode in the true posterior (See Appendix A). Due to this alignment, the solution of the PF-ODE with $X_t = x_t$ approximates the mode with the highest density in the true posterior distribution.

When $\sigma_t^2$ is not sufficiently small, we can still consider other conditions under which the PF-ODE will converge to the closest center $\mu^*$. Consider the five-Gaussian Mixture Model (GMM) example in Figure 3 with the initial point $X_t = x_t$. An intuitive approach is to observe that the normal plane of the velocity $v_t$ divides the space into two regions. In one region, the velocity has a negative projection in the direction of some centers, while in the other region, the velocity has a positive projection in the direction of other centers. Consequently, $X_t$ will move away from centers with negative projections and towards centers with positive projections. Among the centers with positive projections, if the closest center $\mu^*$ also forms the smallest angle with the velocity vector, it is very likely that the PF-ODE will eventually converge to this closest center.

Even if the solution of PF-ODE does not converge to the highest density mode of posterior, we can still show that its density has a non-zero lowerbound with high probability independent of $d$:

**Proposition 3.3.** *The solution of the PF-ODE has a positive likelihood in the true posterior with high probability. More precisely, with probability $1 - e^{-0.132d}$ (w.r.t. the randomness of $x_t \sim p(X_t)$), the following lower bound holds:*

$$p(X_0 = \Phi(t, x_t)|X_t = x_t) \geq \frac{1}{N} \exp\left(-\frac{2c^2}{\sigma_t^2}\right). \quad (11)$$

### 3.3 A TOY EXAMPLE

To better understand the results discussed above, we provide a toy example in $\mathbb{R}^2$. As illustrated in Figure 4, the source distribution $p(X_0)$ is modeled as a 5-Gaussian Mixture Model (GMM). Each Gaussian component is diagonal with a standard deviation of $\sigma_0 = 0.1$. The centers of the Gaussians are located at $(-1, -1)$, $(-1, 1)$, $(1, 1)$, $(1, -1)$, and $(0, 0)$. We adopt the VE diffusion model with

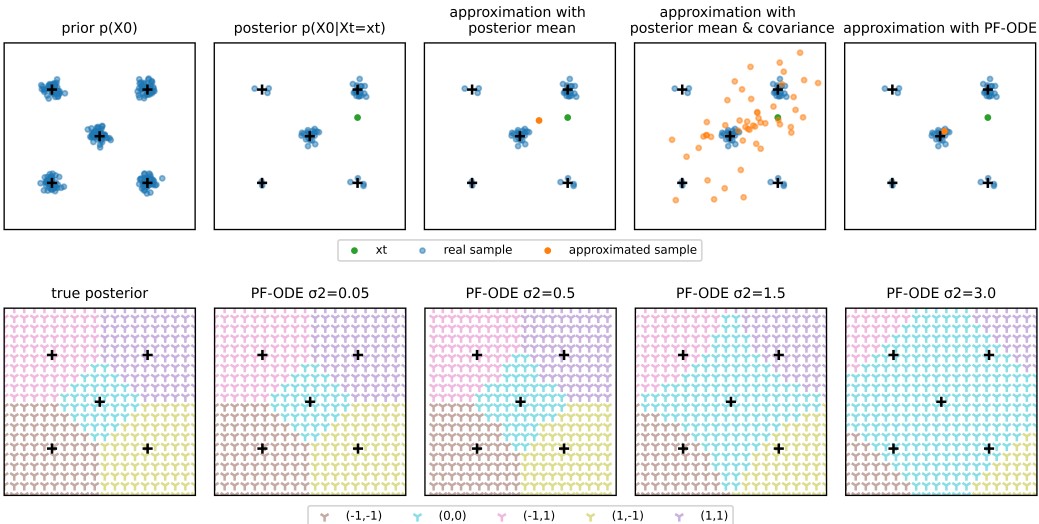

Figure 4: A toy example with 5-GMM.

$\sigma_T = 4$, $T = 100$, and use the $\sigma$ schedule proposed by Karras et al. (2022). The score function $\nabla_{X_t} \log p(X_t)$ is computed analytically.

As shown in Figure 4 (upper part), starting from $x_t = (1, 0.4)$, the posterior $p(X_0|X_t = x_t)$ has most of its density concentrated on the two rightmost Gaussians. However, the approximation using the posterior mean is close to $(1, 0.4)$, a region where the true posterior has almost no density. Additionally, the approximation using both the posterior mean and covariance also concentrates on a region with minimal true posterior density.

We also visualize the decision boundary of the highest density mode for both the true posterior and the PF-ODE starting at different $\sigma_t^2$ values. As shown in Figure 4 (lower part), the decision boundary for the true posterior is always a Voronoi cell centered at each $\mu^i$. When $\sigma_t^2$ is small, the solution of the PF-ODE closely resembles the true posterior. As $\sigma_t^2$ increases, the solution of the PF-ODE starts to deviate from that of the true posterior. Nonetheless, during this phase, the density becomes more evenly distributed in the true posterior. Consequently, the solution of the PF-ODE still maintains a non-zero density.

### 3.4 IMPLEMENTATION OF PF-ODE APPROXIMATION USING CONSISTENCY MODELS

Directly solving the PF-ODE is also computationally intractable for DIS. Fortunately, the PF-ODE can be distilled into a Consistency Model (CM) (Song et al., 2023c). Specifically, CM trains a one-step neural function $g_\theta(t, x_t)$ to approximate the solution of the PF-ODE, $\Phi(t, x_t)$. Its gradient is cheap to evaluate. Thus, we can directly replace the step $x_{0|t} \sim p_\theta(X_0|X_t = x_t)$ in Eq. 5 with $x_{0|t} = g_\theta(t, x_t)$. As this approach is an improvement of diffusion posterior sampling (DPS) (Chung et al., 2022a) using CM, we name it DPS-CM.

In practice, we find that the Consistency Model (CM) often overfits the operator $f(.)$. Specifically, the CM-approximated sample $x_{0|t}$ is close to $y$ after being processed by the operator $f(.)$. However, upon visual inspection, $x_{0|t}$ often appears misaligned with $y$. (See an example in Figure 6). In fact, the resulting overfitted sample is an adversarial example (Szegedy et al., 2013), which aligns with the label $y$ according to the neural network but not according to human perception. To make $f(.)$ more robust, we propose adding small Gaussian noise to the output of the CM, as suggested in the literature on adversarial robustness (Li et al., 2019).

$$x_{0|t} = g_\theta(t, x_t) + \mathcal{N}(0, \tau^2 I). \tag{12}$$

Furthermore, we can also avoid overfitted sample by running CM for multiple steps (Song et al., 2023c). For example, for CM with $K = 2$ steps, we have:

$$x_{0|t} = g_\theta(\tau, g_\theta(t, x_t) + \mathcal{N}(0, \tau^2 I)). \tag{13}$$

To wrap up, we present the final algorithm of DPS-CM in Algorithm 2. For ease of comparison, we also provide the pseudocode of DPS in Algorithm 1 and the difference is highlighted in brown.

| **Algorithm 1:** DPS (Chung et al., 2022a) | **Algorithm 2:** DPS-CM |
|---|---|
| 1 **input** $T, \sigma_t, f(.), y, \Delta(.,.), \zeta_t$ | 1 **input** $T, \sigma_t, f(.), y, \Delta(.,.), \zeta_t, g_\theta(.,.), \tau, K$ |
| 2    $x_T = \mathcal{N}(0, \sigma_T^2 I)$ | 2    $x_T = \mathcal{N}(0, \sigma_T^2 I)$ |
| 3    **for** $t = T$ **to** 1 **do** | 3    **for** $t = T$ **to** 1 **do** |
| 4      $x_{t-1} \sim p_\theta(X_{t-1}|X_t = x_t)$ | 4      $x_{t-1} \sim p_\theta(X_t|X_{t-1} = x_{t-1})$ |
| 5      $x_{0|t} = \mathbb{E}[X_0|X_t = x_t]$ | 5      $x_{0|t} = g_\theta(t, x_t)$ |
| 6      $x_{t-1} \leftarrow x_{t-1} - \zeta_t \nabla_{x_t} \Delta(f(x_{0|t}), y)$ | 6      **for** $k = 1$ **to** $K$ **do** |
| 7    **return** $x_0$ | 7        $x_{0|t} = x_{0|t} + \mathcal{N}(0, \tau^2 I)$ |
| | 8        **if** $K = 1$ **then** |
| | 9          **break** |
| | 10        $x_{0|t} = g_\theta(\tau, x_{0|t})$ |
| | 11      $x_{t-1} \leftarrow x_{t-1} - \zeta_t \nabla_{x_t} \Delta(f(x_{0|t}), y)$ |
| | 12    **return** $x_0$ |

## 3.5 EXTENSION TO LATENT DIFFUSION

DPS-CM can also be extended to diffusion models in latent space in a way similar to how DPS is extended to latent space. Specifically, latent diffusion models (Rombach et al., 2022) employ an encoder $\mathcal{E}(.)$ to map the image into latent space and a decoder $\mathcal{D}(.)$ to map the latent representation back to the image space. Rout et al. (2024) extend DPS into latent DPS by decoding the posterior mean $x_{0|t}$ in **Algorithm 1** before passing it to $f(.)$ during the DPS update:

$$x_{t-1} \leftarrow x_{t-1} - \zeta_t \nabla_{x_t} \Delta(f(\mathcal{D}(x_{0|t})), y). \tag{14}$$

With this simple adaptation, we can run DPS with powerful latent diffusion models such as Stable Diffusion. To extend DPS-CM to latent diffusion, we can additionally replace the consistency model in Algorithm. 2 by latent consistency model (LCM) (Luo et al., 2023b). The resulting algorithm can be described as replacing line 11 of Algorithm. 2 with Eq. 14.

## 4 EXPERIMENTS

### 4.1 EXPERIMENT SETUP

**Base Diffusion Models** For the pixel space diffusion model, we use the pre-trained the Elucidated Diffusion Model (EDM) (Karras et al., 2022)—as provided by Song et al. (2023c). For EDM-related methods, we adopt an ancestral sampler with 1000 Euler steps. For the Consistency Model (CM) (Song et al., 2023c), we employ the official pre-trained model as provided by Song et al. (2023c). For latent diffusion model Rombach et al. (2022), we use Stable Diffusion 1.5 (SD1.5) with Dreamshaper v7 pre-trained weights. For latent consistency model (Luo et al., 2023a), we adopt the official pre-trained model. The detailed setup is described in Appendix B.

**Operators** We evaluate all the methods with four different neural network measurement operators: semantic segmentation, room layout estimation, image captioning, and image classification. For layout estimation, we adopt the neural network proposed by Lin et al. (2018). For semantic segmentation, we use the network developed by Zhou et al. (2017). For image captioning, we employ the BLIP model (Li et al., 2022). For image classification, we use ResNet (He et al., 2015). Additionally, we also evaluate the methods with a non-neural network operator: down-sampling (x4).

**Datasets & Metrics** Following the evaluation protocols of Song et al. (2023c) and Chung et al. (2022a), for EDM, we use the first 1000 images from the LSUN Bedroom and LSUN Cat datasets (Yu et al., 2015) as the test set. All images are resized to 256 by the short edge and cropped into $256 \times 256$. For SD1.5, we use the first 100 images from the LSUN Bedroom and LSUN Cat datasets (Yu et al., 2015) as the test set. Similarly, all images are resized to 512 by the short edge and cropped into $512 \times 512$. We also tried general images such as ImageNet, but the generation is not successful for our complex neural network operator without content specific prompt.

Table 2: Results on neural network measurement operators, *i.e.*, layout estimation, segmentation, caption and classification. **Bold**: best in diffusion-based DIS.

| | LSUN Bedroom | | | | | | | | | LSUN Cat | | |
| --- | --- | --- | --- | --- | --- | --- | --- | --- | --- | --- | --- | --- |
| | Segmentation | | | Layout | | | Caption | | | Classification | | |
| | mIOU | FID | KID | mIOU | FID | KID | CLIP | FID | KID | Acc | FID | KID |
| *Pixel Space (EDM)* | | | | | | | | | | | | |
| DPS | 0.27 | 22.84 | 1.0e-2 | 0.54 | 7.59 | 1.9e-3 | 22.57 | 9.49 | 2.6-e3 | 0.79 | 15.73 | 7.1e-3 |
| FreeDOM | 0.27 | 21.90 | 9.1-e3 | 0.46 | 15.27 | 8.1e-3 | 22.61 | 28.30 | 1.8e-2 | 0.84 | 32.32 | 1.7-e2 |
| MPGD | 0.24 | 82.66 | 6.9e-2 | 0.73 | 15.38 | 8.5-e3 | 21.49 | 21.14 | 1.3e-2 | 0.37 | 15.40 | 7.1e-3 |
| UGD | 0.30 | 13.54 | 4.2e-2 | 0.51 | 7.89 | **1.7e-3** | 22.12 | 9.12 | 2.7e-3 | 0.75 | 14.43 | 7.2e-3 |
| LGD | 0.22 | 35.69 | 2.4e-2 | 0.70 | 8.07 | 2.3e-3 | 22.58 | 8.38 | 2.6-e3 | 0.64 | 13.35 | 4.5e-3 |
| STSL | 0.27 | 19.48 | 7.4e-3 | 0.52 | 7.74 | 2.2e-3 | 22.39 | 9.70 | 2.8e-3 | 0.78 | 15.74 | 6.9e-3 |
| DPS-CM (K=1) | **0.34** | 18.06 | 8.2e-3 | **0.78** | **7.50** | 2.2e-3 | **22.63** | **8.16** | **2.5-e3** | 0.90 | 13.45 | 3.6e-3 |
| DPS-CM (K=2) | 0.31 | **10.14** | **3.7e-3** | 0.76 | 8.12 | 2.7e-3 | 22.34 | 8.80 | 2.8e-3 | **0.93** | **13.30** | **1.7e-3** |
| *Latent Space (Stable Diffusion)* | | | | | | | | | | | | |
| LDPS | 0.28 | 134.1 | 6.4e-2 | 0.40 | 97.12 | 3.6e-2 | 17.23 | 109.8 | 3.9e-2 | 0.76 | 131.2 | 4.1e-2 |
| LGD | 0.29 | 111.7 | 4.5e-2 | 0.50 | 99.15 | 3.6e-2 | **17.25** | 96.92 | 3.4e-2 | **0.92** | 131.3 | 4.5e-2 |
| DPS-CM (K=1) | **0.33** | **104.8** | **4.4e-2** | **0.58** | **93.94** | **3.2e-2** | 17.18 | **90.14** | **2.9e-2** | 0.85 | **127.1** | **4.0e-2** |

Table 3: The posterior sample approximation of different methods.

Table 4: Temporal and spatial complexity of different methods.

| | Approximation of posterior sample | Valid image? |
| --- | --- | --- |
| DPS | $x_{0\|t} = \mathbb{E}[X_0\|X_t]$ | ✗ |
| FreeDOM | $x_{0\|t} = \mathbb{E}[X_0\|X_t]$ | ✗ |
| MDPG | $x_{0\|t} = \mathbb{E}[X_0\|X_t]$ | ✗ |
| UGD | $x_{0\|t} = \mathbb{E}[X_0\|X_t]$ | ✗ |
| LGD | $x_{0\|t} \sim \mathcal{N}(\mathbb{E}[X_0\|X_t], r_t^2 I)$ | ✗ |
| STSL | $x_{0\|t} \sim \mathcal{N}(\mathbb{E}[X_0\|X_t], \mathrm{Cov}(X_0\|X_t))$ | ✗ |
| DPS-CM | $x_{0\|t} = g_\theta(t, X_t)$ | ✓ |

| | Time (s) | VRAM (GB) |
| --- | --- | --- |
| DPS | 150 | 5.35 |
| DPS-CM (K=1) | 218 | 6.32 |
| DPS-CM (K=2) | 340 | 10.2 |

To evaluate sample quality, we employ the Fréchet Inception Distance (FID) (Heusel et al., 2017) and Kernel Inception Distance (KID) (Binkowski et al., 2018). To evaluate the consistency with the constraint, we use mIOU for segmentation and layout, CLIP score for captioning, and accuracy for classification. For neural network measurement operators $f(.)$, we use different models for DIS and testing to avoid over-fitting (see Appendix B). For down-sampling, we use image restoration metrics such as LPIPS (Zhang et al., 2018) and Peak Signal-to-Noise Ratio (PSNR).

**Previous State-of-the-Art DIS** We compare DPS-CM with previously published DIS methods that can handle neural network measurement operators $f(.)$. For methods that directly use the posterior mean as the posterior sample, we include DPS (Chung et al., 2022b), FreeDOM Yu et al. (2023), MPGD (He et al., 2024) and UGD (Bansal et al., 2023). For methods that construct an approximated posterior distribution with the posterior mean as the mode, we include LGD (Song et al., 2023b) and STSL (Rout et al., 2023) (see Table 3). We implement all these methods with the EDM and Euler ancestral sampler (see details in Appendix B). We acknowledge that there are other very competitive works designed for linear operators or those that do not provide open-source implementations (Chung et al., 2023; Song et al., 2023a; He et al., 2023b; Rout et al., 2024; Meng & Kabashima, 2022; Dou & Song, 2023; Chung et al., 2022b; Song et al., 2022; Boys et al., 2023; He et al., 2023a). However, our focus is currently on neural network measurement operators $f(.)$, and therefore, we have not included these methods for comparison.

## 4.2 MAIN RESULTS

**Results on neural network measurement operators** We evaluate our DPS-CM on four neural network measurement operators: segmentation, layout estimation, image captioning, and classification. As shown in Table 2, Figure 1, Figure 5, Figure 11, and Figure 12, both quantitatively and visually, our DPS-CM (Sec. 3.4) achieves significant improvements in both consistency (e.g., mIOU) and sample quality (e.g., FID) compared to the baseline DPS (Chung et al., 2022a). The superiority of

our approximation over other posterior-mean-based approximations is clear, which is attributed to the sensitivity of neural network measurement operators $f(.)$ to out-of-distribution inputs.

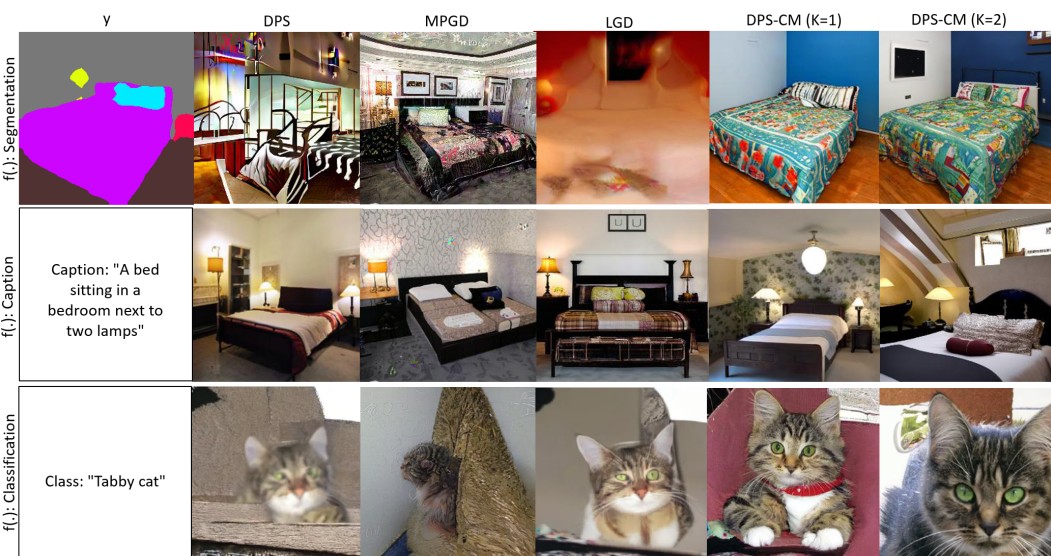

Figure 5: Visual results on neural network measurement operators such as segmentation, caption and classification.

**Results on Non-neural network measurement operators** In addition to neural network measurement operators, we also verify that DPS-CMes work well for non-neural network measurement operators such as down-sampling. Results are summarized in Table 7 and Figure 6 (lower part). Our DPS-CM performs comparably to DPS for simple operators. This is because non-neural network measurement operators $f(.)$ are not as sensitive to out-of-distribution approximations.

**Result on Latent Diffusion** In addition to pixel diffusion (EDM), we also evaluate our DPS-CM on latent diffusion, using Stable Diffusion and latent consistency model. As shown in Table 2 and Table 9, when operator $f(.)$ is neural network, DPS-CM outperforms DPS in latent space. And when operator $f(.)$ is simple linear operators, DPS-CM is comparable to DPS in latent space.

### 4.3 ABLATION STUDY

We evaluate the effect of using CM for posterior approximation in our DPS-CM (Sec. 3.4) and adding randomness to CM in Table 5. More specifically, we demonstrate that using CM to replace the posterior mean reduces the distance to the measurement $y$ in terms of mIOU and improves sample quality as measured by FID. Additionally, adding randomness further enhances mIOU and reduces FID.

Table 5: Ablation study of DPS-CM. **Bold**: Method with best performance.

| | CM | K | Rand | mIOU | FID |
|---|---|---|---|---|---|
| | ✗ | - | ✗ | 0.27 | 22.84 |
| DPS-CM | ✓ | 1 | ✗ | 0.31 | 19.29 |
| | ✓ | 1 | ✓ | **0.34** | 18.06 |
| | ✓ | 2 | ✓ | 0.31 | **10.14** |

Table 6: Ablation study of randomness and data augmentation with segmentation.

| Rand | mIOU | | |
|---|---|---|---|
| | Model A | Model A + DA | Model B |
| ✗ | 0.57 | 0.43 | 0.31 |
| ✓ | 0.51 | 0.54 | 0.34 |

We hypothesize that CM benefits from added randomness because it avoids overfitting the operator $f(.)$, or it enhances $f(.)$'s robustness to adversarial examples (Li et al., 2019). To test this hypothesis, we use Model A for $f(.)$ during DIS training. During evaluation, we compare the results using Model A, Model A with data augmentation (DA), and a separate Model B. In Table 6, we show that when tested with Model A, DIS without randomness outperforms DIS with randomness. However,

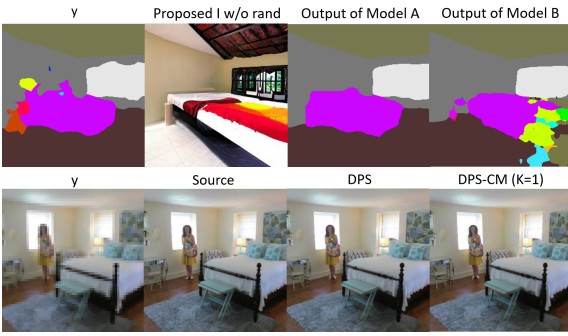

Figure 6: *upper.* An example of over-fitting an operator $f(.)$. *lower.* Visual results on linear operators such as down-sampling.

Table 7: Results on non-neural network measurement operators such as super-resolution.

|  | Bedroom Down-sampling (x4/x8) | | | |
|---|---|---|---|---|
|  | KID | FID | LPIPS | PSNR |
| *Pixel Space (EDM)* | | | | |
| DPS | 1.4e-3 | 4.82 | **0.11** | 26.69 |
| FreeDOM | **1.1e-3** | 4.84 | **0.11** | 26.70 |
| MPGD | 2.0e-3 | 43.36 | 0.38 | 22.84 |
| LGD | 2.8e-3 | 7.10 | 0.14 | 26.51 |
| STSL | 1.2e-3 | **4.80** | 0.12 | 26.65 |
| DPS-CM (K=1) | 1.9e-3 | 5.67 | 0.12 | **26.91** |
| *Latent Space (Stable Diffusion)* | | | | |
| LDPS | **1.9e-2** | 77.36 | **0.27** | 28.63 |
| LGD | 5.1e-2 | 107.2 | 0.30 | 28.72 |
| DPS-CM (K=1) | 2.8e-2 | 86.66 | **0.27** | **28.98** |

when tested with Model A with DA and Model B, DIS with randomness outperforms DIS without randomness. This indicates that DIS without randomness overfits Model A. An example of such overfitting is presented in Figure 6 (upper part).

## 4.4 COMPLEXITY

In Table 4, we show that our DPS-CM has comparable temporal and spatial complexity to DPS (Chung et al., 2022a). DPS-CM with $K = 1$ brings around 25% more running time and 20% more memory consumption. While DPS-CM with $K = 2$ takes around twice more time and memory.

## 5 RELATED WORK

An important branch of DIS focuses on linear operators $f(.)$ using projection or pseudo-inverse techniques (Wang et al., 2022; Kawar et al., 2022; Chung et al., 2022b; Lugmayr et al., 2022; Song et al., 2022; Dou & Song, 2023; Pokle et al., 2024; Cardoso et al., 2024). For general, non-linear $f(.)$, various approaches have been proposed, including Monte Carlo methods (Wu et al., 2024a; Phillips et al., 2024), proximal gradient methods (Xu & Chi, 2024), and variational inference (Feng et al., 2023; Mardani et al., 2023; Janati et al., 2024). Among these paradigms, conditional score estimation methods are the most widely adopted as they are scalable to practically large images with reasonable runtime (Chung et al., 2022a; Yu et al., 2023; Zhu et al., 2023; He et al., 2024; Song et al., 2023b; Boys et al., 2023; Rout et al., 2023; 2024; Bansal et al., 2023). Following this paradigm, we propose to approximate the posterior sample using the PF-ODE, which improves results for neural network measurement operators $f(.)$. After the submission of this paper, we notice that Wang et al. (2024); Wu et al. (2024b) also solve the DIS for general noisy operator. And Zhao et al. (2024) also leverage CM to solve DIS.

## 6 DISCUSSION & CONCLUSION

One limitation of this paper is that we only consider simple image datasets such as LSUN Bedroom and LSUN Cat. Training-free generation with conditions such as segmentation can be very challenging on complicated image datasets like ImageNet. Consequently, most previous works have focused on simpler images, such as faces and dogs (Yu et al., 2023; Bansal et al., 2023). Therefore, we also focus on simple image datasets and leave the generation of more complex images as future work.

To conclude, we show that the solution of PF-ODE is an effective posterior sample. Built upon this, we propose to use CM as a high-quality approximation to posterior sample. Further, we propose a new family of DIS using only CM. Experimental results show that our DPS-CM perform well for neural network measurement operators.

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

# A PROOF OF MAIN RESULTS

We first derive some basic properties of the GMM model. More specifically, we have

$$p(X_0) = \frac{1}{N} \sum_{i=1}^{N} \mathcal{N}(X_0 | \mu^i, \sigma^2 I), \tag{15}$$

$$p(X_t | X_0) = \mathcal{N}(X_t | X_0, \sigma_t^2 I), \tag{16}$$

$$p(X_t) = \frac{1}{N} \sum_{i=1}^{N} \mathcal{N}(X_t | \mu^i, (\sigma^2 + \sigma_t^2) I), \tag{17}$$

$$p(X_0 | X_t) = \frac{p(X_t | X_0) p(X_0)}{p(X_t)} = \sum_{i=1}^{N} \frac{p(X_t | X_0)}{\sum_{j=1}^{N} \mathcal{N}(X_t | \mu^j, (\sigma^2 + \sigma_t^2) I)} \mathcal{N}(X_0 | \mu^i, \sigma^2 I)$$

$$= \sum_{i=1}^{N} u_i \mathcal{N}(X_0 | \mu^i, \sigma^2 I),$$

$$\text{where } u_i = (\exp - \frac{\|X_0 - X_t\|^2}{2\sigma_t^2}) / (\sum_{j=1}^{N} \frac{1}{\sqrt{(1 + \sigma^2/\sigma_t^2)^d}} \exp(-\frac{\|X_t - \mu^i\|^2}{2(\sigma^2 + \sigma_t^2)})). \tag{18}$$

For true posterior $p(X_0 | X_t = x_t)$, we know that $X_0$ eventually converges to the vicinity of one of the $\mu^i$ with high probability (for small $\sigma$). Therefore, the $\mu^*$ closest to $X_t$ has highest weighting $u_i$. Thus no matter what is the value of $\sigma_t^2$, the highest density mode of true posterior is always the mode $\mu^*$ that is closest to the initial point $x_t$. The decision boundary is always the Voronoi diagram centered at $\mu^i$.

**Lemma 3.3** *The PF-ODE can be written as:*

$$\frac{dX_t}{dt} = \underbrace{\sum_{i=1}^{N} \frac{w_i}{2} \frac{d\sigma_t^2}{dt} \frac{(X_t - \mu^i)}{\sigma^2 + \sigma_t^2}}_{\text{velocity field } v_t}, w_i = (\exp(-\frac{\|X_t - \mu^i\|^2}{2(\sigma^2 + \sigma_t^2)})) / (\sum_{j=1}^{N} \exp(-\frac{\|X_t - \mu^j\|^2}{2(\sigma^2 + \sigma_t^2)})).$$

*Proof.* We need to compute the score function first:

$$\nabla \log p(X_t = x_t) = \frac{\nabla p(X_t = x_t)}{p(X_t = x_t)}$$

$$= \frac{1}{p(X_t = x_t)} \nabla (\sum_{i=1}^{N} \frac{1}{N} (\mathcal{N}(X_t = x_t \mid \mu^i, (\sigma^2 + \sigma_t^2) I)))$$

$$= \frac{1}{p(X_t = x_t)} \sum_{i=1}^{N} \frac{1}{N} \mathcal{N}(X_t = x_t \mid \mu^i, (\sigma^2 + \sigma_t^2))(-\frac{(x - \mu^i)}{\sigma^2 + \sigma_t^2})$$

$$= \sum_{i=1}^{N} ((\exp(-\frac{\|x_t - \mu^i\|^2}{2(\sigma^2 + \sigma_t^2)})) / (\sum_{j=1}^{N} \exp(-\frac{\|x_t - \mu^j\|^2}{2(\sigma^2 + \sigma_t^2)})))(-\frac{x - \mu^i}{\sigma^2 + \sigma_t^2}). \tag{19}$$

Combining with the PF-ODE in Eq. 3, we can obtain the result. $\qquad\square$

With those basic properties, we can show that the solution of PF-ODE with initial value $X_t = x_t$ has non-zero density in true posterior $p(X_0 | X_t)$. Prior to that, we need an extra lemma:

**Lemma A.1.** *As PF-ODE is margin preserving, the solution of PF-ODE concentrate inside a ball centered at $\mu^i$ with radius $\sqrt{d}\sigma$ i.e., $\exists k, s.t. \|\Phi(t, x_t) - \mu^k\|^2 \le 2d\sigma^2$ with probability $1 - e^{-0.134d}$ $\forall x_t \sim p(X_t)$.*

*Proof.* We can use the Proposition 4.3 of Yang et al. (2024), which states that for a $d$-dimensional diagonal Gaussian variable $x$,

$$Pr(\|x - \mu\|^2 \geq d\sigma^2 + 2d\sigma^2(\epsilon + \sqrt{\epsilon})) \leq e^{-d\epsilon}. \tag{20}$$

Then we can simply let $\epsilon + \sqrt{\epsilon} = \frac{1}{2}$, which leads to the choice $\epsilon = 0.134$. $\qquad\square$

**Proposition 3.2** *The solution of the PF-ODE has a positive likelihood in the true posterior with high probability. More precisely, with probability $1 - e^{-0.132d}$ (w.r.t. the randomness of $x_t \sim p(X_t)$), the following lower bound holds:*

$$p(X_0 = \Phi(t, x_t)|X_t = x_t) \geq \frac{1}{N} \exp\left(-\frac{2c^2}{\sigma_t^2}\right). \tag{21}$$

*Proof.*

$$
\begin{aligned}
p(X_0 = \Phi(t, x_t) \mid X_t = x_t) &= \frac{p(X_t|X_0)p(X_0)}{p(X_t)} \\
&= \frac{p(X_t|X_0)p(X_0)}{\sum_{X_t} p(X_t|X_0)p(X_0)} \\
&\overset{(a)}{=} \sum_{i=1}^{N} \frac{p(X_t|X_0)}{\sum_{j=1}^{N} \mathcal{N}(X_t|\mu^j, (\sigma^2 + \sigma_t^2)I)} \mathcal{N}(X_0|\mu^i, \sigma^2 I) \\
&= \sum_{i=1}^{N} u_i \mathcal{N}(X_0 = \Phi(t, x_t)|\mu^i, \sigma^2 I) \\
&\geq u_j \mathcal{N}(X_0 = \Phi(t, x_t)|\mu^j, \sigma^2 I), \forall j
\end{aligned}
$$

where (a) holds due to Eq 18. We let $k = \min_i\{\|\mu^i - \Phi(t, x_t)\|\}$, by Lemma. A.1, we have $\|\Phi(t, x_t) - \mu^k\| \leq 2d\sigma^2$ with probability $1 - e^{-0.132d}$.

$$
\begin{aligned}
p(X_0 = \Phi(t, x_t)|X_t = x_t) &\geq u_k \frac{1}{\sqrt{(2\pi\sigma^2)^d}} \exp\left(-\frac{\|\mu^k - \Phi(t, x_t)\|^2}{2\sigma^2}\right) \\
&\overset{(a)}{\geq} u_k \frac{1}{\sqrt{(2\pi\sigma^2)^d}} \exp\left(-\frac{2d\sigma^2}{2\sigma^2}\right) \\
&= \frac{\exp\left(-\frac{\|\Phi(t, x_t) - x_t\|^2}{2\sigma_t^2}\right)}{\sum_{j=1}^{N} \sqrt{(1 + \sigma^2/\sigma_t^2)^d} \exp\left(-\frac{\|x_t - \mu^i\|^2}{2(\sigma^2 + \sigma_t^2)}\right)} \frac{1}{\sqrt{(2\pi\sigma^2)^d}} \exp\left(-d\right) \\
&\overset{(b)}{\geq} \frac{\exp\left(-\frac{4c^2}{2\sigma_t^2}\right)}{\sqrt{2^d} \sum_{j=1}^{N} \exp\left(0\right)} \frac{1}{\sqrt{(2\pi\sigma^2)^d}} \exp\left(-d\right) \\
&= \frac{1}{N} \frac{1}{\sqrt{(4\pi\sigma^2)^d}} \exp\left(-\frac{2c^2}{\sigma_t^2} - d\right) \\
&= \frac{1}{N} \exp\left(\left(\log \frac{1}{\sqrt{4\pi\sigma^2}}\right)d - d - \frac{2c^2}{\sigma_t^2}\right)
\end{aligned}
$$

(a) holds due to Lemma. A.1. (b) holds due to $\|\Phi(t, x_t) - x_t\| \leq \|\Phi(t, x_t)\| + \|x_t\|$. As they are both bounded by $c$, $\|\Phi(t, x_t) - x_t\|^2$ is bounded by $4c^2$. And $1 + \frac{\sigma^2}{\sigma_t^2} \leq 2$. Then, we can set a small enough $\sigma$, such as

$$\sigma < \frac{1}{\sqrt{4\pi e}}, \log \frac{1}{\sqrt{4\pi\sigma^2}} > 1. \tag{22}$$

Then we have

$$p(X_0 = \Phi(t, x_t)|X_t = x_t) \geq \frac{1}{N} \exp\left(-\frac{2c^2}{\sigma_t^2}\right) > 0, \tag{23}$$

which completes the proof. $\qquad\square$

We can study the PF-ODE in Eq. 10 informally when $\sigma_t^2$ is rather small or large. When $\sigma_t^2$ is small, the soft-max $w_i$ becomes a "hard"-max. Denote $k = \min_i\{\|\mu^i - \Phi(t, x_t)\|\}$ and the PF-ODE at that time can be written as

$$\frac{dX_t}{dt} = -\frac{1}{2}\frac{d\sigma_t^2}{dt}\left(-\frac{(X_t - \mu^k)}{\sigma^2 + \sigma_t^2}\right). \tag{24}$$

At that time, the PF-ODE is first order separable. And we can solve it with initial value $X_t = x_t$ as

$$\frac{\Phi(t, x_t) - \mu^k}{x_t - \mu^k} = e^{h(t)}, \tag{25}$$

where $h(.)$ is some function of $t$ related to the $d\sigma_t^2/dt$.

Let's assume a simple $\sigma_t^2$ schedule such as $\sigma_t^2 = t^2$. In that case, we have

$$\Phi(t, x_t) = \mu^k + (x_t - \mu^k)e^{\frac{1}{2}\ln\frac{\sigma^2}{\sigma^2 + t^2}}. \tag{26}$$

The solution has the form of $\mu^k$ with an offset term weighted by an exponential term. When $\sigma^2$ is small, the exponential term goes to 0 very fast. And therefore $\Phi(t, x_t) \approx \mu^k$ at that time.

# B  ADDITIONAL EXPERIMENT SETUP

## B.1  IMPLEMENTATION DETAILS

All the experiments are implemented in Pytorch, and run in a computer with AMD EPYC 7742 CPU and Nvidia A100 GPU.

Table 8: The model specification used for different non-linear operators.

|  | Model A | Model B |
|---|---|---|
| Segmentation | MobileNet + C1 | ResNet50 + PPM |
| Layout | Lin et al. (2018) | Lin et al. (2018) + DA |
| Caption | BLIP | CLIP |
| Classification | ResNet50 | VITB16 |

As we have shown, using the same model for $f(.)$ causes overfitting for neural network based $f(.)$. Therefore, we adopt different model for $f(.)$ in DIS and testing, and the details are shown in Table 8.

Table 9: Metrics for DIS loss and evaluation.

|  | $d(.,.)$ for DIS | metric for Test |
|---|---|---|
| Segmentation | Cross Entropy | mIOU |
| Layout | Cross Entropy | mIOU |
| Caption | Cross Entropy | CLIP score |
| Classification | Cross Entropy | Accuracy |
| Downsample | MSE | MSE |

For different operators, we also have different $d(.,.)$ to evaluate the distance $d(f(x_{0|t}), y)$ during the DIS process. For all four non-linear operators, the cross entropy are used for $d(.,.)$. While for down-sample, we adopt MSE. To evaluate how consistent the generated samples are to $y$, we use y-metrics. Or to say, the metrics computed with input measurement $y$ and $f(x_{0|t})$. More specifically, for Segmentation and Layout, we evaluate consistency by y-mIOU. For image caption, we evaluate consistency by CLIP score (Hessel et al., 2021). For classification, we evaluate consistency by accuracy. And for down-sample, we evaluate consistency by MSE. Note that the $d(.,.)$ used during DIS follows the convention of training corresponding $f(.)$, and the y-metric used for testing also follows the convention of testing corresponding $f(.)$.

For latent diffusion, we set the prompt to be "A high quality image of a bedroom" for LSUN bedroom dataset, and "A high quality image of a cat" for the LSUN cat dataset.

## B.2 DETAILS OF DIS ALGORITHM

Below we provide the detailed algorithm of different DIS methods including the ones we compare to and our own.

---

**Algorithm 3:** FreeDOM (Yu et al., 2023)

1 **procedure**
   FreeDOM($p_\theta(.|.), q(.|.), T, f(.), y, d(.,.), \zeta_t, r, K$)
2      $x_T = \mathcal{N}(0, T^2 I)$
3      **for** $t = T$ **to** 1 **do**
4        **for** $t' = K$ **to** 1 **do**
5         $x_{t-1} \sim p_\theta(X_{t-1}|X_t = x_t)$
6         $x_{0|t} = \mathbb{E}[X_0|X_t = x_t]$
7         $x_{t-1} \leftarrow x_{t-1} - \zeta_t \nabla_{x_t} \Delta(f(x_{0|t}), y)$
8         **if** $t' \neq 1, t \in r$ **then**
9          $x_t = q(X_t|X_{t-1} = x_{t-1})$
10        **else**
11         **break**
12     **return** $x_0$

**Algorithm 4:** MPGD (He et al., 2024)

1 **procedure**
   MPGD($q_\theta(.|.,.), T, f(.), y, d(.,.), \zeta_t$)
2      $x_T = \mathcal{N}(0, T^2 I)$
3      **for** $t = T$ **to** 1 **do**
4        $x_{0|t} = \mathbb{E}[X_0|X_t = x_t]$
5        $x_{0|t} \leftarrow x_{0|t} - \zeta_t \nabla_{x_t} \Delta(f(x_{0|t}), y)$
6        $x_{t-1} \leftarrow q(X_{t-1}|X_t = x_t, X_t = x_{0|t})$
7      **return** $x_0$

---

**Algorithm 5:** UGD (Bansal et al., 2023)

1 **input** $T, \sigma_t, f(.), y, \Delta(.,.), \zeta_t, \eta_t$
2      $x_T = \mathcal{N}(0, \sigma_T^2 I)$
3      **for** $t = T$ **to** 1 **do**
4        $x_{t-1} \sim p_\theta(X_{t-1}|X_t = x_t)$
5        $x_{0|t} = \mathbb{E}[X_0|X_t = x_t]$
6        $x'_{0|t} = x_{0|t} - \eta_t \nabla_{X_{0|t}} \Delta(f(x_{0|t}), y)$
7        $x_{t-1} \leftarrow x_{t-1} - \zeta_t \nabla_{x_t} \Delta(f(x_{0|t}), y)$
8        $x_{t-1} \leftarrow x_{t-1} + (x'_{0|t} - x_{0|t})$
9      **return** $x_0$

**Algorithm 6:** LGD (Song et al., 2023b)

1 **procedure**
   LGD($p_\theta(.|.), T, f(.), y, d(.,.), \zeta_t, r_t$)
2      $x_T = \mathcal{N}(0, T^2 I)$
3      **for** $t = T$ **to** 1 **do**
4        $x_{t-1} \sim p_\theta(X_{t-1}|X_t = x_t)$
5        $x_{0|t} = \mathbb{E}[X_0|X_t = x_t] + \mathcal{N}(0, r_t^2 I)$
6        $x_{t-1} \leftarrow x_{t-1} - \zeta_t \nabla_{x_t} \Delta(f(x_{0|t}), y)$
7      **return** $x_0$

---

**Algorithm 7:** STSL (Rout et al., 2023)

1 **procedure**
   STSL($p_\theta(.|.), T, f(.), y, d(.,.), \zeta_t, \eta_t$)
2      $x_T = \mathcal{N}(0, T^2 I)$
3      **for** $t = T$ **to** 1 **do**
4        $x_{0|t} = \mathbb{E}[X_0|X_t = x_t]$
5        $x_t \leftarrow x_t - \zeta_t \nabla_{x_t} \Delta(f(x_{0|t}), y)$
6        $\epsilon \sim \mathcal{N}(0, I)$
7        $x_t \leftarrow$
   $x_t - \eta_t \nabla_{x_t}(\epsilon^T(s_\theta(t, x_t + \epsilon) - s_\theta(t, x_t)))$
8        $x_{t-1} \sim p_\theta(X_{t-1}|X_t = x_t)$
9      **return** $x_0$

---

**FreeDOM** Yu et al. (2023) propose to adopt the time-travel that is designed specifically for inpainting (Lugmayr et al., 2022) to general operator $f(.)$. (See Algorithm. 3) More specifically, it proposes an inner loop that goes forward after a backward step with forward kernel $q(.|.)$. The new hyper-parameters are time-travel steps $K$ and time-travel range $r$.

**MPGD** He et al. (2024) propose to perform the gradient ascent directly on posterior mean instead of on $x_t$. And the posterior mean after gradient ascent is used to correct the score function (See Algorithm. 4). They claim that their approach is able to converge faster and outperform DPS when $T = 20, 100$. However, as we use $T = 1000$, the advantage of their approach is not clearly shown in our experiments.

**UGD** Bansal et al. (2023) further improve DPS by introducing a backward guidance. More specifically, after the DPS step, UGD additionally performs $K$ MPGD update step, and optimizes the posterior mean directly,

**LGD** Song et al. (2023b) adopt a Gaussian approximation to posterior sample (See Algorithm. 6). More specifically, they use an additive of Gaussian noise and posterior mean as an approximation of Gaussian sample. And the mean of approximated sample is the same as real posterior sample. This approach is later improved by Boys et al. (2023); Rout et al. (2023) to second order. Or to say, they estimate the covariance of Gaussian using second order Tweedie's formula. And the mean and covariance of approximated sample is the same as real posterior sample. The authors of LGD further propose a multi-sample approach to reduce gradient variance. However, we only use LGD with sample size $n = 1$ for fair comparison. The new hyper-parameters is variance $\tau$.

**STSL** Rout et al. (2023) propose to improve Song et al. (2023b) by estimating the posterior as Gaussian distribution with posterior mean and posterior covariance. As directly estimating the posterior covariance using second order Tweedie's formula is expensive, they propose a Monte Carlo estimation and the resulting algorithm is shown in Algorithm. 7. However, we only use STSL with sample size $n = 1$ for fair comparison.

### B.3 HYPER-PARAMETERS

| | Segmentation | Layout | Caption | Classification | Down-sampling |
|---|---|---|---|---|---|
| DPS | $\zeta = 256.0$ | $\zeta = 7.2$ | $\zeta = 24.0$ | $\zeta = 8.0$ | $\zeta = 14.4$ |
| LGD | $\zeta = 256.0$ | $\zeta = 7.2$ | $\zeta = 24.0$ | $\zeta = 8.0$ | $\zeta = 14.4$ |
| | | | $K = 1, \tau = 0.2$ | | |
| FreeDOM | $\zeta = 256.0$ | $\zeta = 7.2$ | $\zeta = 24.0$ | $\zeta = 8.0$ | $\zeta = 14.4$ |
| | | | $K = 2, r = [100, 200]$ | | |
| MPGD | $\zeta = 2560.0$ | $\zeta = 72.0$ | $\zeta = 240.0$ | $\zeta = 80.0$ | $\zeta = 144.0$ |
| Proposed | $\zeta = 256.0$ | $\zeta = 7.2$ | $\zeta = 24.0$ | $\zeta = 8.0$ | $\zeta = 14.4$ |
| | | | $\tau = 0.2$ | | |

Table 10: The hyper-parameters of other DIS and DPS-CM.

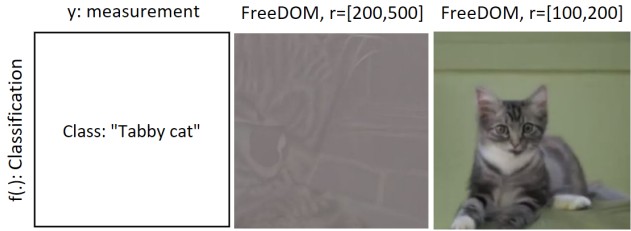

Figure 7: Comparison of different inner-loop range for FreeDOM.

We list the detailed hyper-parameters of DPS, LGD, FreeDOM, MPGD (Chung et al., 2022a; Song et al., 2023b; Yu et al., 2023; He et al., 2024) and two of our DPS-CM in Table 10. For all the methods, one common hyper-parameter is the step size $\zeta$ used in gradient descent. For LGD and our DPS-CM, an additional hyper-parameter is the additional additive noise $\tau = 0.2$. We do not use the multi-sample LGD as it is significantly slower than all other approaches. For FreeDOM, the additional parameters are time-travel steps $K$, and time-travel range $r$. We set $K = 2$ for fair comparison, as a large $K$ make FreeDOM significantly slower than all other approaches. We set $r = [100, 200]$ instead of $r = [200, 500]$ in original paper (Yu et al., 2023). This is because we find that setting $r = [200, 500]$ in VE-diffusion has significant negative effect on sample quality. The visual comparison is in Figure 7.

### B.4 DETAILS OF TABLE 1

In Table 1, we compare the sample quality of different posterior sample approximation methods. More specifically, we adopt 1000 step EDM (Karras et al., 2022). We start from $t = 900$ and attempt to sample from $p(X_0|X_900)$. We compare posterior mean, posterior mean with Gaussian and the result of CM (PF-ODE) by FID and KID.

## C ADDITIONAL EXPERIMENT RESULTS

### C.1 ADDITIONAL VISUAL RESULTS

We present more visual results for non-linear operators in Figure 11 and 12. We can see that our DPS-CM has the best consistency with measurement $y$ and the best sample quality for most instances.

We present more visual results of down-sampling in Figure 8. It is shown that our DPS-CM works as good as DPS.

We present visual results of latent diffusion in Figure 9. It is shown that our DPS-CM also works well in latent space.

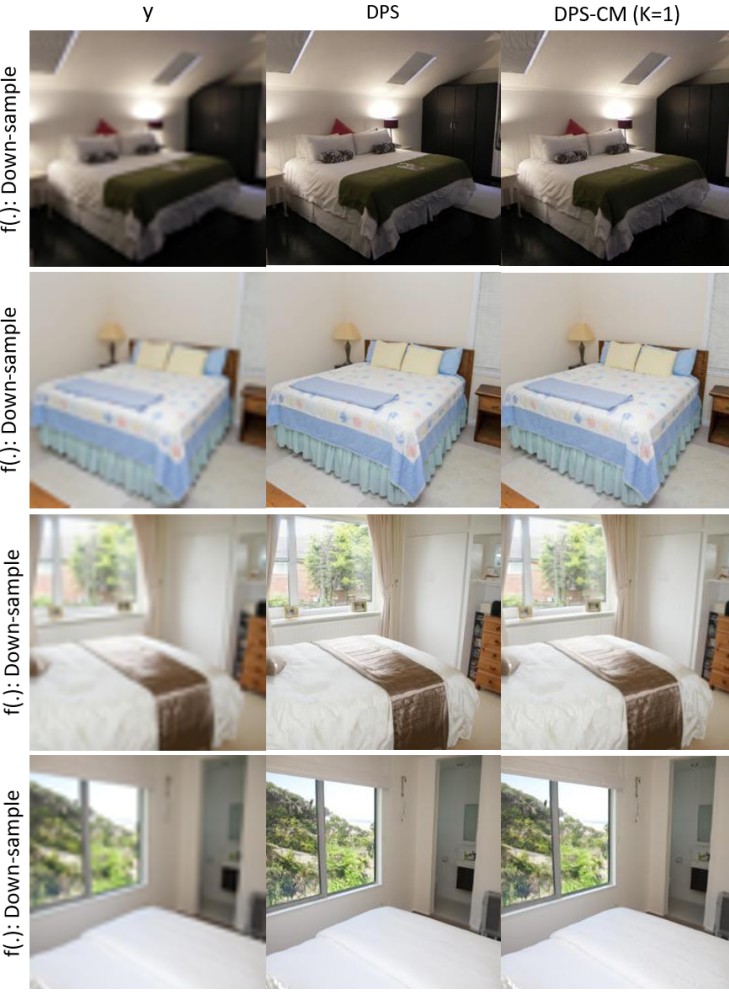

Figure 8: Additional visual results on image down-sampling.

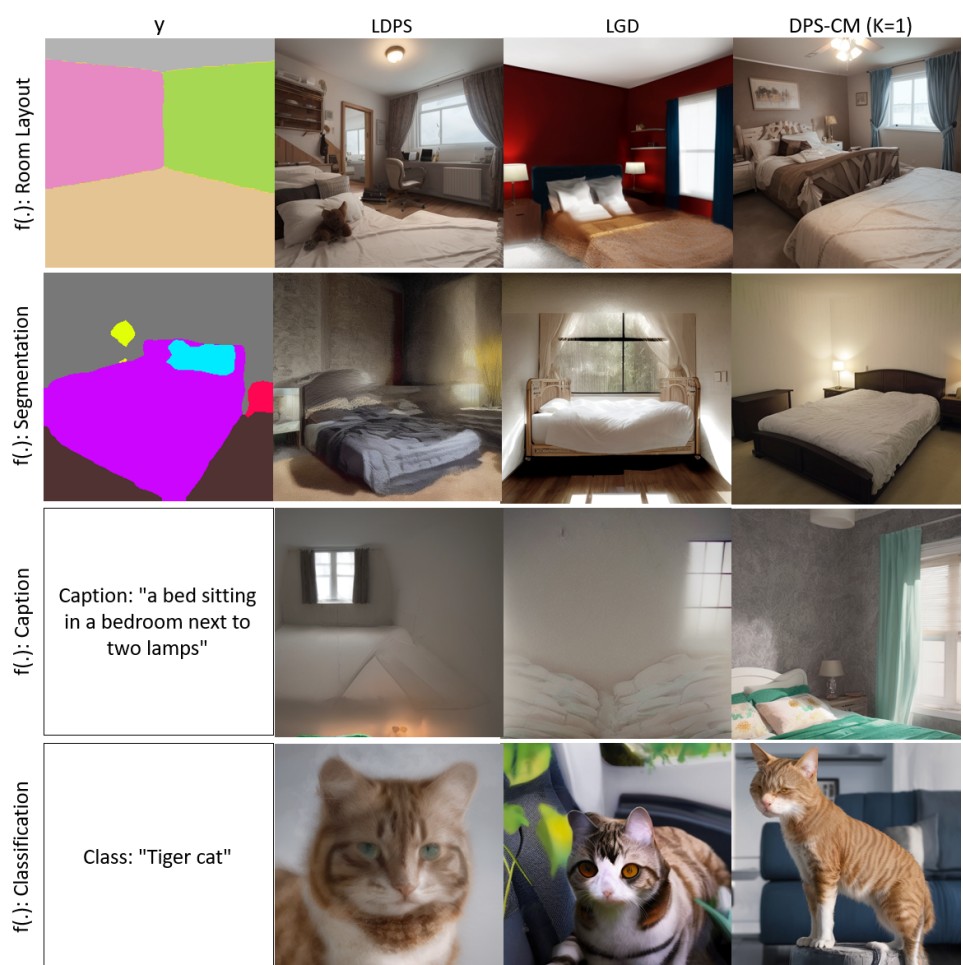

Figure 9: Additional visual results of latent diffusion.

## C.2 FAILURE CASES

When the measurement $y$ is too far from diffusion prior, DPS-CMes and other DIS approaches fail. An example of such failure is shown in Figure 10. The input measurement $y$ describes a woman. However, human is not a part of LSUN bedroom dataset. And none-of the DIS approaches is able to generate a woman. And the samples generated by DIS look like unconditional sample.

## D ADDITIONAL DISCUSSION

### D.1 REPRODUCIBILITY STATEMENT

The proof of all theoretical results are shown in Appendix. A. For experiments, all two datasets are publicly available. In Appendix. B, we provide additional implementation details of all other DIS that we compare to. Further, detailed hyper-parameters of all baselines and our DPS-CM are presented. Besides, we provide source code for reproducing empirical results as supplementary material.

### D.2 BROADER IMPACT

The approach proposed in this paper allows conditional generation without training a new model. This saves the energy of training conditional generative diffusion model and reduces the carbon

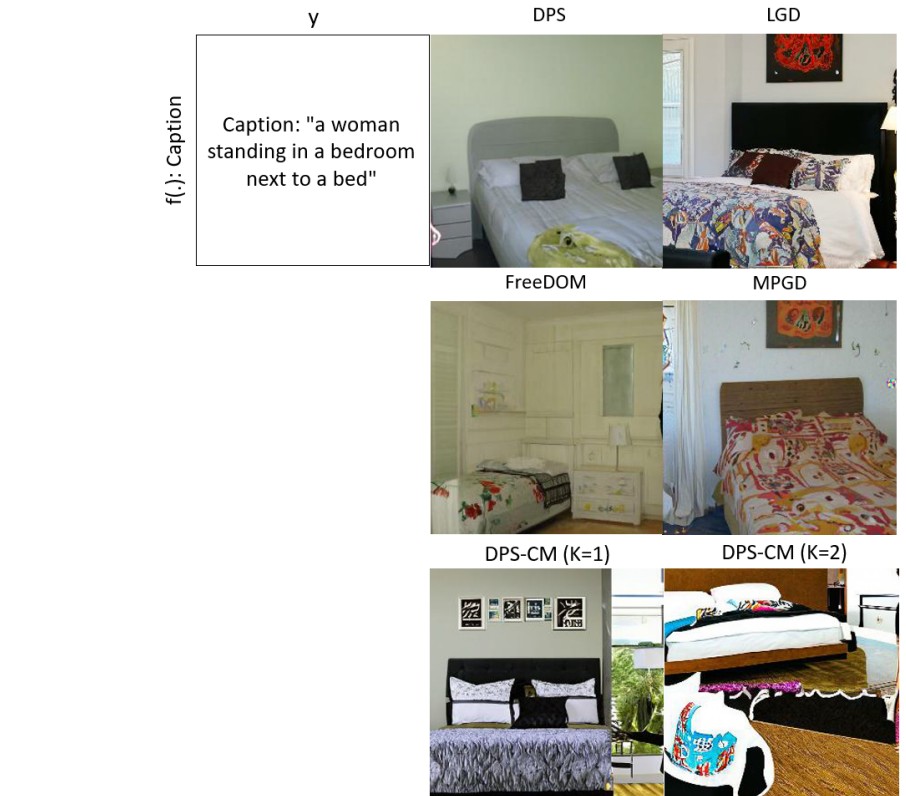

Figure 10: Visual results of a failure case.

emission. Potential negative impact is the same as other conditional generative model, such as trustworthiness brought by generating fake image.

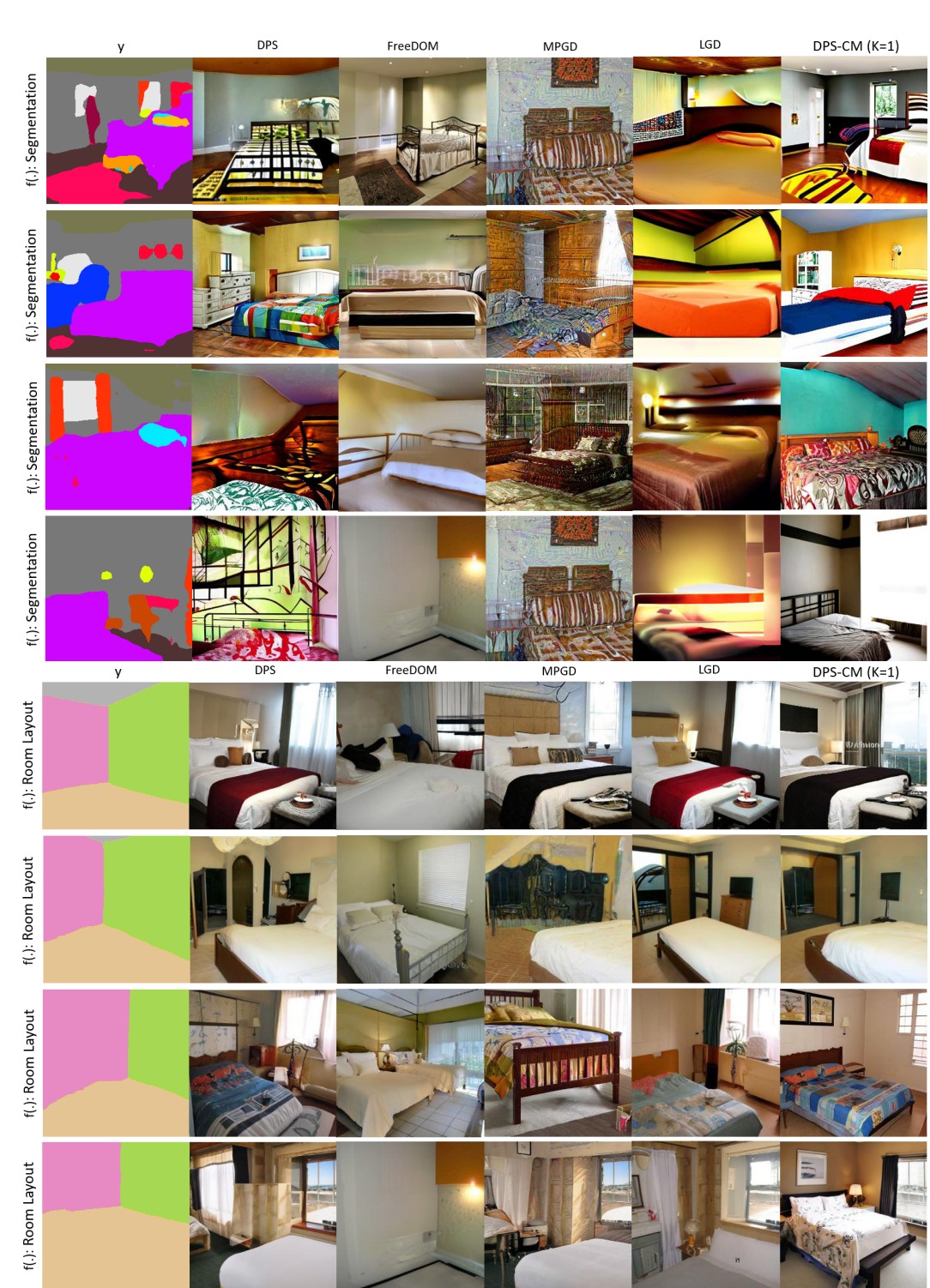

Figure 11: Additional visual results on image segmentation and layout estimation.

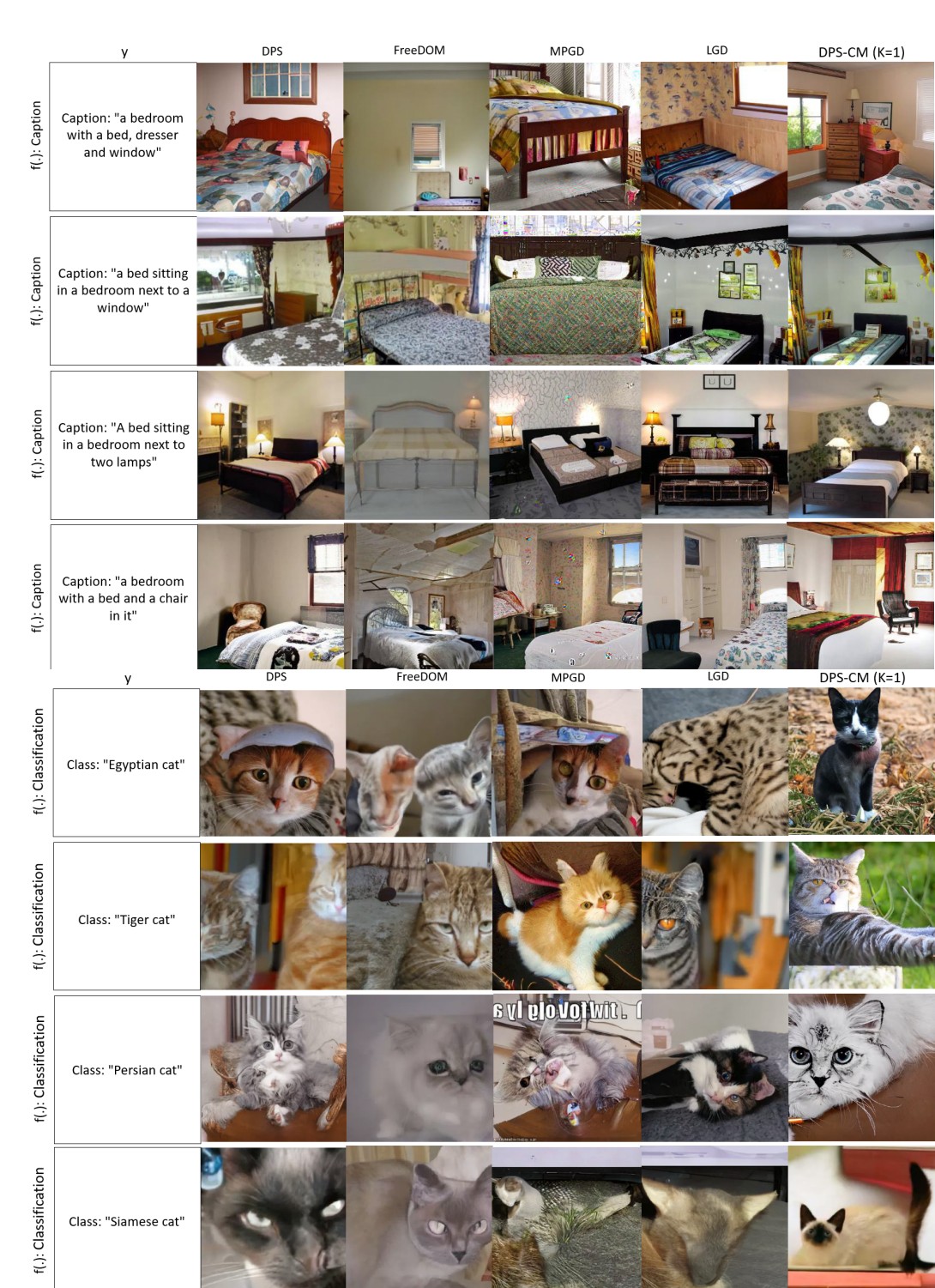

Figure 12: Additional visual results on image captioning and image classification.

