# OpenReview forum: "Consistency Model is an Effective Posterior Sample Approximation for Diffusion Inverse Solvers"
_ICLR.cc/2025/Conference — Submitted to ICLR 2025_

### Official Review · Reviewer_LTYd · 2024-11-03

**Soundness:** 3
**Presentation:** 2
**Contribution:** 2
**Rating:** 6
**Confidence:** 5

**Summary:**

The authors propose a new approach using CMs to generate realistic image samples in Diffusion Inverse Solvers, improving the application of complex, non-linear neural network operators like those in  semantic segmentation, room layout estimation, image captioning, and image classification. Unlike traditional methods that produce low probability images, the incorporation of CMs is expected to maintain sample realism, resulting in more accurate posterior approximations, particularly when neural network-based operators are involved.

**Strengths:**

The idea of using CMs in the process of inverse problems/posterior approximation seems novel and interesting.

The proposed method outperforms baseline approaches, particularly when compared to straightforward extension

**Weaknesses:**

Although being interesting, the novelty of the presented work is marginally above acceptation threshold since the only contribution seems to be the use of CMs in order to compute $x_0$ from $x_t$.

The manuscript could benefit from improved clarity and organization, as certain sections are challenging to follow. See further remarks.

Further remarks:

In the presented algorithm, the updates are stated as $\zeta_t \Delta\left(f(x_0 \mid t), y\right)$, where $\Delta$ is defined as some distance. The update of $x_t$ however should be the gradient of that distance.

In order to enhance the readability of the work, the authors should think about introducing a clear distinction between the posterior $p(x_0|x_t)$ and the posterior $p(x|y)$, given y is the observation.

The abbreviation DPS (Diffusion posterior sampling) is used but never introduced.

**Questions:**

The presented manuscript often states likelihood terms as $p_\theta(y|x)$. Please elaborate on why it contains theta, as the likelihood in general is not a learned function with the same parameters as the learned data distribution $p_\theta(x)$.

Furthermore, I would be interested as to which extend the PF-ODE solution differs from the MAP solution?

---

> ### Author Response · Authors · 2024-11-23
>
> Thank you for your detailed review. We are pleased to provide answers to your questions:
>
>
> ## W1 The manuscript could benefit from improved clarity and organization
> Thanks a lot for the detailed comments and suggestions.
> We have fixed the typos in the revised version of the paper.
>
> * The $\zeta_t\Delta(f(\mathbb{E}[X_0|X_t]), y)$ should be the gradient of that distance: We changed it into $\zeta_t\nabla\Delta(f(\mathbb{E}[X_0|X_t]), y)$.
> * The authors should think about introducing a clear distinction between the $p(X_0|X_t)$ and $p(X|y)$: We have fixed it by calling the $p(X_0|X_t)$ posterior and $p(X|y)$ conditional distribution.
> * The abbreviation DPS is used but never introduced: we defined it in page 6 upon its first appearance.
>
> ## Q1 why $p_{\theta}(y|X_0)$ contains $\theta$?
> * Thanks for pointing it out. $p(y|X_0)$ is independent of $\theta$, and hence there should be no subscript. We have fixed it in the revised version.

---

> > ### Comment · Reviewer_LTYd · 2024-11-29
> >
> > Thank you for your response and addressing my comments. I will keep my rating.

---

### Official Review · Reviewer_cxtg · 2024-11-03

**Soundness:** 3
**Presentation:** 3
**Contribution:** 3
**Rating:** 5
**Confidence:** 4

**Summary:**

This paper is about solving inverse problems using a pre-trained denoising diffusion prior. Posterior sampling with diffusion models requires an estimate of the score $\nabla _{x_t} \log p_t(x_t) + \nabla _{x_t} \log \int p(y | x_0) p _{0|t}(x_0 | x_t) d x_0$. While the first term is  estimated using the pre-trained score, the second term is usually very difficult to estimate accurately. A common approximation used in the literature involves using $\nabla _{x_t} \log p(y | E[X_0 | X_t = x_t])$ where  $E[X_0 | X_t = x_t]$ can also be estimated using the pre-trained score via Tweedie's formula. This approximation results in many efficiencies well documented in the literature and this paper tries to circumvent them using by using a sample from the PF-ODE as a replacement. Specifically, the authors use consistency models to speed up the process of sampling from the PF-ODE and ensuring that the differentiation is not costly.

**Strengths:**

The idea is quite original wrt to the literature. The paper is quite illustrated and clear. The experiments are interesting and extensive; the authors compare to many existing methods, on both pixel space and latent space diffusion.

**Weaknesses:**

- The most obvious weakness of the method is the use of consistency models since these are quite difficult to train and pre-trained CMs are not widely available.

- In my opinion the justification for the method is rather weak. The paper argues that using a sample from the PF-ODE is valid because the sample has zero density and that furthermore, for the Gaussian mixture example considered, the PF-ODE sample has non-zero density under the posterior $p(x_0 | x_t)$ with high probability. At the same time it is also easy to find examples of a likelihood function $p(y|x_0)$ such $\int p(y|x_0) p(x_0 | x_t) d x_0 > 0$ for all $x_t$ but $p(y|\Phi(t, x_t)) = 0$ for $x_t$ in a set of positive Lebesgue measure. As a result, I'm not totally convinced that the argument is strong.

**Questions:**

The presentation of Proposition 3.3 is a bit misleading. The authors should state in the assumption the condition on $\sigma$ that they use in the proof in order to get a lowerbound independent of the dimension, or remove this claim after the proposition.

---

> ### Author Response · Authors · 2024-11-23
>
> Thanks for your detailed comments. It appears that there have been some misunderstandings, and we have provided our responses to your questions below.
>
> ## W1 consistency models since these are quite difficult to train and pre-trained CMs are not widely available
> CM is one of the most widely adopted approach to distill diffusion models. Training large scaled CM has been studied extensively (see e.g., [Simplifying, Stabilizing and Scaling Continuous-Time Consistency Models] [Latent Consistency Models: Synthesizing High-Resolution Images with Few-Step Inference]), and currently there are several high quality CMs that are easily accessible online:
> * CM for ImageNet 64, LSUN Cat and Bedroom 256: https://github.com/openai/consistency_models
> * CM for Stable Diffusion 1.5, SD-XL and PixArt alpha: https://github.com/luosiallen/latent-consistency-model
>
> ## W2 the justification for the method is rather weak
> In DPS, we only care about whether the approximated sample follows $p(X_0|X_t)$ or not. As long as the samples follows $p(X_0|X_t)$, DPS estimate the conditional score perfectly (Eq 5), and thus achieve sampling from $p(X_0|y)$. The $p(y|X_0)$ part in $p(y|X_t)$ will be handled by gradient ascent of DPS. Even if $p(y|X_0)=0$ for some initial $t$ and $X_0$, the gradient ascent will make it large for subsequent $t$.
>
> ## Q1 The presentation of Proposition 3.3 is a bit misleading
> Thanks for the comments. We have included $\sigma < \frac{1}{\sqrt{4 \pi e}}$ in the assumption.

---

> > ### Comment · Reviewer_cxtg · 2024-11-26
> >
> > Thank you for your response and for addressing my concerns. I still think that the methodology has limited applicability; requiring the user to train both a diffusion model and a consistency model is a non-negligible constraint is not justifiable given that the proposed method does not offer significant performance gains, relative to the significant computational cost it introduces. I am keeping my initial score.

---

### Official Review · Reviewer_kz1a · 2024-11-04

**Soundness:** 3
**Presentation:** 3
**Contribution:** 3
**Rating:** 5
**Confidence:** 3

**Summary:**

This paper presents an interesting approach for posterior sampling using $p(x_0|x_t)$ being approximated via consistency model. Results show improvement over baselines such as DPS

**Strengths:**

1. Easy to follow
2. The idea of using CM to approximate the PF-ODE solution is interesting

**Weaknesses:**

1. Using CM over the existing inverse problem solving adds a lot of computational burden, while the benefit is not clearly visible. Even though the $x_0|x_t$ may be off the manifold of ground truth image distribution, this does not imply a sacrifice in reconstruction quality as demonstrated in this work [1]. The quantitative results do not show a significant improvement over DPS.
2. CM itself can be used as a good prior for solving inverse problems: see this work [2]. This paper needs to compare with more recent works in inverse problem solving
3. More DIS baselines are desired such as DDNM [3]




[1]. Wang, Hengkang, et al. "DMPlug: A Plug-in Method for Solving Inverse Problems with Diffusion Models." arXiv preprint arXiv:2405.16749 (2024). NeurIPS 2024

[2]. Zhao, Jiankun, Bowen Song, and Liyue Shen. "CoSIGN: Few-Step Guidance of ConSIstency Model to Solve General INverse Problems." ECCV 2024

[3]. Wang, Yinhuai, Jiwen Yu, and Jian Zhang. "Zero-Shot Image Restoration Using Denoising Diffusion Null-Space Model." The Eleventh International Conference on Learning Representations.

**Questions:**

1. More DIS baselines are desired such as CoSIGN [2], DDNM [3]

I am open to change my rating if authors could address my concerns (how clearly a CM model could benefits with inverse problem solving comparing to strong baselines like DDNM)

---

> ### Author Response · Authors · 2024-11-23
>
> Thank you for your detailed review. We are pleased to provide answers to your questions.
> We hope that our response can clarify the misunderstanding and we kindly ask you to reassess our contribution.
>
> ## W1 Using CM over the existing inverse problem solving adds a lot of computational burden
> We would like to clarify that the additional time and space complexity introduced by using CM is fairly small and should not be considered a significant burden. In Table 4 we have shown that using CM increase time complexity by 30% and increase RAM usage by 1GB. In Table 3 and Figure 5, it is shown that using CM improve DPS a lot, both quantatively and qualitatively.
>
> The mentioned paper [DMPlug: A Plug-in Method for Solving Inverse Problems with Diffusion Models] is a concurrent work, which will not be published formally until Dec 2024. We were unable to use the insights from a paper published after ICLR deadline (and the citation of it is not required by ICLR policy).
>
> ## W2 CM itself can be used as a good prior for solving inverse problems
> [CoSIGN: Few-Step Guidance of ConSIstency Model to Solve General INverse Problems] is a concurrent paper published in Oct 2024. So we could not compare with it before the ICLR deadline. Besides, CoSIGN requires training, while our approach, DPS and other methods that we compare to, are zero-shot. In our opinion, CoSIGN and our approach attempt to solve different problems under different setting, hence should not be compared directly. However, its insight or idea may be helpful in zero-shot setting and futher exploration is an interesting future direction.
>
> ## W3 More DIS baselines are desired such as DDNM:
> Approaches such as DDNM [Zero-Shot Image Restoration Using Denoising Diffusion Null-Space Model] require $f(.)$ to be linear. While in this paper, we consider more general and complex scenarios where $f(.)$ may be a neural network. To the best of our knowledge, DDNM cannot be used to solve non-linear problems. We have already included six strong baselines such as DPS, FreDOM, MPGD, UGD, LGD, STSL and clearly demonstrated the advantage of our method over the baselines when $f(.)$ is a neural network.

---

> > ### Comment · Reviewer_kz1a · 2024-11-25
> > **Thanks for the response**
> >
> > Thanks authors for their response. In my sense, the benefits of using an additional CM is still not justified both empirically and theoretically, I will maintain my original rating.

---

### Official Review · Reviewer_h6fG · 2024-11-04

**Soundness:** 2
**Presentation:** 2
**Contribution:** 2
**Rating:** 5
**Confidence:** 3

**Summary:**

The paper proposes using consistency models (CMs) to solve inverse problems with diffusion models. In diffusion inverse problems, common approaches need to go from xt to x0 at every iteration to be able to compute a measurement-guided gradient with respect to the measurement y from x0. The majority uses the expectation from Tweedie's formula to compute x0 from tx, which may not result in a good example for complex multi-modal distribution. The paper proposes to replace this step with a CM. They show improvement upon prior work.

**Strengths:**

The paper considers an important problem: how diffusion models can be used to solve complex inverse problems, such as giving a segmented image to reproduce the underlying image. The paper acknowledges a known limitation on one assumption that prior work makes on the distribution of the data (which allows to use the Tweedie's formula), and proposes to address it by consistency models.

**Weaknesses:**

I find the contribution of the contribution incremental, and the presentation can be improved. The main weaknesses are as follows: the mathematical formulation is confusing and the approach to training the CM is not fully fair compared to the baselines.

1. The paper trains a CM to be used for diffusion-based inverse solvers. It's not clear why such mathematical details (some are not fully proper) are included, which I do not think is the main contribution of the paper. It's unclear why in (1) the authors formulate the Markov chain following a VE formulation. Could the author explain this choice? The conventional Markov chain for diffusion models is the popular one based on VP, which has a different mean and variance such that the structure is destroyed, but the energy of the process remains the same.

2. Solving inverse problems with diffusion mostly occurs in the regime where the diffusion model is trained unconditionally without the knowledge of the measurement operator f(.) (see the DPS used for vision problems such as deblurring). However, the paper in Section 3.4 discusses that the CM is overfitting to f(.) and proposes an approach to make the framework robust. So this brings the following: the CM going from xt to x0 seems to be trained with the knowledge of the measurement. If f(.) is involved during the training, then the trained framework is not general anymore (it's problem specific). Hence, the comparison of the proposed framework to models such as DPS, where the model is not trained based on the measurement operator, is not fair. Please provide more information and clarity if this is not the case. The fair comparison would be a scenario where both methods are trained a similar condition (e.g., not having the knowledge of the forward operator).

Here are some questions concerning this:

- Is CM trained with knowledge of f(.), or if this is a misunderstanding?
- If the CM is trained with f(.), please explain and justify comparing it to methods like DPS that don't use this information?

More comments are below:


Lack of thorough literature

- The limitations of DIS are not fully explained in the intro. Indeed, the mean-based approximation is one challenge. A few others are related to whether methods such as DPS are doing posterior sampling or using the measurement to guide the process onto likely solutions (see [1]).


The paper needs improvement in presentation. Here are a few examples

- While the notations such as Xt and X0 are known to the reader with knowledge of diffusion models, these are used in the abstract and intro without introducing them. Hence, I suggest re-writing the abstract without these notations and introducing the diffusion in the introduction before using x0, xt, etc.

- Consistency models are not defined and introduced in the intro, but the authors explain that they are used to improve performance. I suggest the authors to provide a brief definition or explanation of consistency models in the introduction.

- How the results are generated for Table 1; this appears abruptly without proper explanation. I suggest to include a brief explanation of the methodology used to generate the results in Table 1.

Some terms within the manuscript are not precise and clear. Please provide clarifications.

- Section 1: with "neural network operators"? Does this refer to the measurement operator or the score function of the diffusion? I suggest to say "measurement operators" instead of "operators". Please clarify "neural network operators"?


[1] Wu, Z., Sun, Y., Chen, Y., Zhang, B., Yue, Y., & Bouman, K. L. (2024). Principled Probabilistic Imaging using Diffusion Models as Plug-and-Play Priors. arXiv preprint arXiv:2405.18782.

**Questions:**

see above

---

> ### Author Response · Authors · 2024-11-23
>
> Thanks for your detailed comments. It appears that there have been some misunderstandings, and we have provided our responses to your questions below.
> We hope that our response can clarify the misunderstanding and you can reassess our contribution.
>
> ## W1 why in (1) the authors formulate the Markov chain following a VE formulation?
>
> The result of this paper is heavily based on stochastic differential equation (SDE) and probability flow ordinary equation (PF-ODE). The classic literature that introduces SDE and PF-ODE adopts VE notation [Score-Based Generative Modeling through Stochastic Differential Equations] and we follow the tradition in this paper. Besides, the VE and VP formulations are essentially equivalent and can be made interchangable by linear scaling. In popular projects such as Huggingface/diffusers, VP and VE diffusions are implementated in a unified way: VE diffusion solvers (Euler) can be applied to solve a VP diffusion (Stable Diffusion 2.0), and vice versa.
>
> ## W2 the paper in Section 3.4 discusses that the CM is overfitting to f(.) and proposes an approach to make the framework robust:
>
> We would like to emphasize here that No CM is trained for $f(.)$. Our approach works for the zero-shot setting, which is the main motivation of the paper.
>
> ## Q1: Is CM trained with knowledge of f(.), or if this is a misunderstanding?
>
> This question is related to the W2, and we would like to clarify that NO new CM is trained with the knowledge of f(.).
> We use existing unconditional CMs which do not know f(.).
>
> ## Q2: If the CM is trained with f(.), please explain and justify comparing it to methods like DPS that don't use this information?
>
> Again, CM is not trained with the knowledge $f(.)$.
>
> ## W3 Lack of thorough literature
>
> Thanks for the pointer. We will cite the reference [Principled Probabilistic Imaging using Diffusion Models as Plug-and-Play Priors] and discuss its relation with our work.
>
> However, we would like to note that reference provided is a concurrent work: it will not be formally published until Dec 2024, which is after the deadline of ICLR 2025. Hence, we think not discussing this paper should not be considered as lack of thorough literature.
>
> ## W4 The paper needs improvement in presentation
>
> Thanks a lot for the comments and suggestion. We have polished the presentation as follows in the revised version:
> * We have removed the notations in the abstract.
> * We have added the definition of consistency models in introduction.
> * We have included the implementation details of Table 1.
> * "neural network operators" means the measurement operators $f(.)$. We have rephrased it as "neural network measurement operators".

---

> > ### Comment · Reviewer_h6fG · 2024-11-25
> > **Reviewer's Response**
> >
> > I thank the authors for their response. I have read all the reviews and responses. My concerns remain. Here are my comments.
> >
> > [Literature]
> >
> > - I would like to clarify that the example paper [1] I cited was intended to highlight that the literature review in this work is not comprehensive. While the authors focus on addressing the mean-based approximation challenge in diffusion models, they overlook other known limitations of “guidance-based” approaches, such as their inability to accurately perform posterior sampling. Paper [1] is one example discussing this issue, and I strongly suggest the authors, now that they have such knowledge, cite it in the related works or introduction to strengthen their paper.
> >
> > [Formulation]
> >
> > - My question regarding the VE formulation was to inquire why the authors include extensive mathematical details about this formulation, especially since it is not their contribution nor directly related to the core aspects of their CM framework.
> >
> > [On  f(.) ]
> >
> > - I thank the authors for clarifying that CM is not trained when f(.)  is present. However, on line 313, the authors mention that CM overfits to the operator f(.) . If CM is not trained conditionally, this statement seems misleading. Could the authors clarify their intended use of the term “overfit” in this context?
> >
> > [New Comment – Framework Comparison]
> >
> > - The authors apply their framework solely to DPS. As noted by another reviewer, the improvement over DPS is marginal. Overall, the paper lacks sufficient comparisons. For instance, how does their framework perform when integrated into other methods provided in the tables or compared against newer diffusion-based inverse solvers such as PSLD [A], ReSample [B], or the general formulation of ReSample (DAPS [C])?
> >
> > [New Comment – Applicability to  x_0|t  Guidance]
> >
> > - The paper evaluates CM only in conjunction with DPS, where the guidance gradient updates  $x_t$ . I wonder how applicable the proposed framework is to cases where guidance updates  $x_{0|t}$ . Many newer diffusion-based inverse solvers, including MPGD and ReSample, employ the latter approach and have demonstrated better performance. Could the authors explore or discuss the applicability of their framework in this context?
> >
> > Overall, as pointed out by another reviewer, the computational complexity of the framework needs a trained model, and the presented results do not justify the practical benefit of this approach.
> >
> > [A] Rout, L., Raoof, N., Daras, G., Caramanis, C., Dimakis, A., & Shakkottai, S. (2024). Solving linear inverse problems provably via posterior sampling with latent diffusion models. Advances in Neural Information Processing Systems, 36.
> >
> > [B] Song, B., Kwon, S. M., Zhang, Z., Hu, X., Qu, Q., & Shen, L. Solving Inverse Problems with Latent Diffusion Models via Hard Data Consistency. In The Twelfth International Conference on Learning Representations.
> >
> > [C] Zhang, B., Chu, W., Berner, J., Meng, C., Anandkumar, A., & Song, Y. (2024). Improving diffusion inverse problem solving with decoupled noise annealing. arXiv preprint arXiv:2407.01521.
> >
> >
> > Typo: Table 3: "MDPG" --> "MPGD"?

---

### Author Response · Authors · 2024-11-23
**Summary of Revisions**

Summary of Revisions

Thank you for your detailed review. We have uploaded the revised paper, with all revisions marked in blue. Below is a summary of the revisions:

* We have removed notation from the abstract, as suggested by h6fG.
* We have rephrased "neural network operators" to "neural network measurement operators," as suggested by h6fG.
* We have specified the $\sigma$ value in the assumption, as suggested by cxtg.
* We have modified $\zeta_t\Delta(f(\mathbb{E}[X_0|X_t]), y)$ to $\zeta_t\nabla_{X_t}\Delta(f(\mathbb{E}[X_0|X_t]), y)$, as suggested by LTYd.
* We have distinguished between $p(X_0|X_t)$ and $p(X|y)$ by referring to the former as the posterior and the latter as the conditional distribution, as suggested by LTYd.
* We have added the definition of the consistency model and diffusion posterior sampling, as suggested by LTYd and h6fG.
* We have removed $\theta$ from $p_{\theta}(y|X_0)$, as suggested by LTYd.
* We have included implementation details for Table 1, as suggested by h6fG.
* We have cited and discussed additional literature, as suggested by h6fG and kz1a.

We hope that we have addressed your concerns, and we are glad to provide additional clarifications if needed.

---

### Meta-Review · Area_Chair_71oA · 2024-12-23

**Metareview:**

The paper explores the use of consistency models (CMs) to address inverse problems in diffusion models by replacing the traditional expectation computation step with a CM-based approach. This allows for faster sampling from the probability flow ODE (PF-ODE) while avoiding costly differentiation steps. The proposed method demonstrates improvements over prior approaches, highlighting an interesting application of CMs in solving inverse problems.

Strength: The paper discusses an interesting question, which tries to address the limitations of existing methods. Although it has been shown in previous works, the idea of leveraging CMs for solving inverse problems is interesting and hasn’t been explored too much in the literature. It may open some interesting directions for diffusion-based inverse problem-solving.

Weakness: Reviewers raise several common concerns, mainly on the justification why the additional CM is necessary in solving the inverse problems. Specifically, The approach introduces computational overhead, as it requires training both a diffusion model and a CM, which could be a significant constraint given the difficulty in training CMs. Additionally, the quantitative results show limited improvement over existing methods like DPS, and the comparison with more recent baselines would be more appreciated. Besides, the mathematical formulation and paper presentation could be further improved.

Overall, although the target problem is interesting, the paper may need a major revision to improve clarity and method justification. Thus, I recommend reject.

**Additional Comments On Reviewer Discussion:**

In the rebuttal, all reviewers have responded to the author’s rebuttal and some also followed up with more questions. But there is only one round of response from the author without following up to address the further concerns. With no more discussion, I think these concerns still remain in the current version.

---

### Decision · Program_Chairs · 2025-01-22

Reject